# CAUSAL CONCEPT GRAPH MODELS: BEYOND CAUSAL OPACITY IN DEEP LEARNING

**Gabriele Dominici**[*]
Università della Svizzera italiana
gabriele.dominici@usi.ch

**Pietro Barbiero**[*]
IBM Research[†]
pietro.barbiero@ibm.com

**Mateo Espinosa Zarlenga**
University of Cambridge
me466@cam.ac.uk

**Alberto Termine**
IDSIA
alberto.termine@idsia.ch

**Martin Gjoreski**
Università della Svizzera italiana
martin.gjoreski@usi.ch

**Giuseppe Marra**
KU Leuven
giuseppe.marra@kuleuven.be

**Marc Langheinrich**
Università della Svizzera italiana
marc.langheinrich@usi.ch

## ABSTRACT

*Causal opacity* denotes the difficulty in understanding the "hidden" *causal structure* underlying the decisions of deep neural network (DNN) models. This leads to the inability to rely on and verify state-of-the-art DNN-based systems, especially in high-stakes scenarios. For this reason, circumventing causal opacity in DNNs represents a key open challenge at the intersection of deep learning, interpretability, and causality. This work addresses this gap by introducing Causal Concept Graph Models (Causal CGMs), a class of interpretable models whose decision-making process is causally transparent by design. Our experiments show that Causal CGMs can: (i) match the generalisation performance of causally opaque models, (ii) enable human-in-the-loop corrections to mispredicted intermediate reasoning steps, boosting not just downstream accuracy after corrections but also the reliability of the explanations provided for specific instances, and (iii) support the analysis of interventional and counterfactual scenarios, thereby improving the model's causal interpretability and supporting the effective verification of its reliability and fairness.

## 1 INTRODUCTION

*Causal opacity* refers to the difficulty in understanding a model's decision-making mechanisms (Termine & Primiero, 2024). Like causal discovery, causal opacity, can be assessed through Pearl's framework of causality (Pearl, 1995), categorizing an agent's causal understanding based on its ability to answer three types of questions: *observational* queries ("what is the relationship between the feature $x$ and the model's output?"), *interventional* queries ("what happens if I fix the feature $x$ to a value $k$?"), and *counterfactual* queries ("what would the model's prediction had been if feature $x$ had taken value $k'$ instead of $k$?"). Unlike causal discovery, causal opacity concerns the decision-making process of a DNN model. Whether a DNN is causally opaque (or transparent) depends on whether users can answer interventional and counterfactual questions that concern the model's inferential behaviour. In this regard, notice that (i) the problem of causal opacity is distinct from the problem of *causal discovery*, which involves detecting causal mechanisms in the real world,

---

[*]Equal contribution
[†]Work conducted while employed at Università della Svizzera italiana.

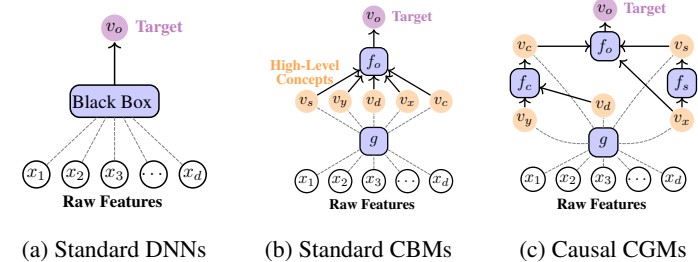

(a) Standard DNNs    (b) Standard CBMs    (c) Causal CGMs

Figure 1: (a) Standard DL models are *black boxes* in the sense that the causal structure of their mapping from raw input features (e.g., pixels of an image) to the target remains *opaque*. (b) In Concept Bottleneck Models (CBM), high-level human-interpretable concepts are first extracted through an encoder $g$ and then used to predict the target. Although CBMs are *semantically transparent*, the causal structure of the model's inference assumes a straightforward causal structure where concepts are causally independent and are all direct causes of the target. (c) In **Causal Concept Graph Models (Causal CGMs)**, both the concepts' *semantics and the inference's causal structure are transparent*.

rather than understanding the causal structure of a model's internal reasoning; and (ii) causal opacity (and the related form of causal understanding) neither implies nor requires discovering the "true" causal structure of the data-generating process. For example, a model might effectively predict heart attacks based on the rule `heart_attack ← smoking ← lung_cancer`, even though the actual biological mechanism is that smoking causes lung cancer, not the reverse. Further details on the distinction between causal opacity and causal discovery are discussed in App. A.

Deep Learning (DL) models (Fig. 1a) are ***causally opaque*** when applied to unstructured data like images, as interventions on low-level features, such as pixels, are not meaningful (Rudin, 2019; Ghorbani et al., 2019) and therefore do not enable us to easily answer any of these sorts of questions. For example, asking "what happens if I fix the intensity of the pixel $x_i$ to $k$?" does little to reveal whether a DNN is exploiting protected attributes like "gender" or "race" when classifying an image of a person as a "doctor". Yet, understanding when these sorts of high-level attributes are misused to make predictions is crucial for the proper verification and vetting of models (Kusner et al., 2017), particularly in high-impact real-world tasks (e.g., medical admission recommendations).

To address this issue, Concept Bottleneck Models (CBMs) (Koh et al., 2020) have been designed to support interventions on high-level interpretable features known as "concepts" (Kim et al., 2018). These concepts, whose semantics are aligned with units of information experts would use to solve the same task, allow for more intuitive interventions and counterfactual analysis (Wachter et al., 2017; Abid et al., 2021). This is because CBMs shift from reasoning based on raw features to explicitly operating on interpretable variables such as "age", "income", "gender", or "BMI".

Although CBMs are ***semantically transparent*** (Kim et al., 2018; Facchini & Termine, 2021)[1], we argue they still rely on **two unrealistic assumptions** (visualised in Fig. 1b). First, they assume that concepts are causally independent, an assumption that breaks in most real-world tasks. For instance, CBMs might treat "smoking" and other health conditions as independent predictors of an insurance premium, ignoring how *intervening on* smoking affects health. Second, and more importantly, CBMs assume concepts are the direct causes of model predictions. These assumptions oversimplify and constrain the model's inference structure, preventing the model from using more complex and effective concept dependencies (Beckers, 2022). As a result, the problem of *causal opacity* still represents a key open challenge at the intersection of DL, causality, and XAI.

**Contributions** In response to this challenge, we introduce *Causal Concept Graph Models* (Causal CGMs, see Figure 1c). Causal CGMs are a new concept-based architecture whose decision-making process is ***causally transparent by design***, avoiding the unrealistic assumption that concepts must be causally independent and direct causes of class prediction. The results of our experiments show that Causal CGMs can: (i) match the generalisation performance of state-of-the-art causally-opaque models, (ii) enable human-in-the-loop corrections of mispredicted intermediate reasoning steps,

---

[1]Here by *semantic transparent* we mean a model that focuses on features that are meaningful and understandable by human experts in the model's application domain.

boosting both downstream accuracy and explanation correctness, and (iii) support the analysis of both interventional and counterfactual questions, improving causal interpretability and enabling effective verification of reliability and fairness. The code related to this paper is publicly available [2].

## 2 BACKGROUND

**Causal Models** A causal model (CM) is a mathematical representation of the rules or laws that explain how different variables of a given target-system causally influence each other. In computer science and statistics, *Structural Causal Models* (SCM) proposed by Pearl (2009) are a key framework for causal modelling. Formally, an SCM $\mathcal{M}$ is a triplet $(\mathcal{U}, \mathcal{V}, \mathcal{F}_\theta)$ where: $\mathcal{U}$ is a set of **exogenous variables** representing latent factors with a causal influence on the modelled target-system; $\mathcal{V}$ is a set of **endogenous variables** representing observable and measurable variables; $\mathcal{F}_\theta$ is a set of functions (or *structural equations*) parametrised by $\theta$ describing the **causal mechanisms**, which determine the values of each endogenous variable $v_i \in \mathcal{V}$ by computing the conditional probability $p(v_i|u_i, \text{pa}(v_i); \theta)$, where $u_i \in \mathcal{U}$ and $\text{pa}(v_i)$ are, respectively, the set of exogenous and endogenous variables causally affecting $v_i \in \mathcal{V}$. Every SCM corresponds to a directed acyclic graph (DAG) whose nodes represent variables and edges represent direct causal connections. Hard interventions in SCMs are modelled through the ***do*-operator** (Pearl, 1995; 2009), which modifies the model's structure by fixing an endogenous variable to a value $\kappa \in \mathbb{R}$, breaking its original causal dependencies. Formally, applying $do(v_i = \kappa)$ to a model $\mathcal{M}$ results in a new model $\mathcal{M}_{v_i=\kappa}$ which is identical to $\mathcal{M}$ except that the equation for $v_i$ is replaced with a constant value $\kappa$, removing all arrows into $v_i$ in the DAG.

**Concept-based models** Concept-based models (Kim et al., 2018; Chen et al., 2020; Yeh et al., 2020; Kim et al., 2023; Barbiero et al., 2023; Oikarinen et al., 2023) are interpretable architectures that explain predictions using high-level information units (i.e., "concepts"). Most follow the Concept Bottleneck Model (CBM) (Koh et al., 2020) approach. Given a sample's raw features $\mathbf{x} \in X \subseteq \mathbb{R}^d$ (e.g., an image's pixels), a set of $r$ concepts $c_i \in C \subseteq \{0, 1\}^r$ (e.g., "red", "round"), and a set of $l$ class labels $y_j \in Y \subseteq \{0, 1\}^l$ (e.g., labels "apple" or "tomato"), a CBM estimates the conditional distribution $\prod_j p(y_j \mid \text{pa}(y_j) = c_1, \ldots, c_r; \theta_f) \prod_i p(c_i \mid \mathbf{x}; \theta_g)$ where $p(c_i \mid \mathbf{x}; \theta_g)$ is a set of independent Bernoulli distributions. This formulation relies on **two key CBMs' assumptions**: i) *concepts directly cause class predictions*, and ii) *concepts are conditionally independent of one another*. Usually, concept-based models represent a concept $c_i$ using its predicted probability $\hat{c}_i \in [0, 1]$. However, this may degrade task accuracy when concepts are incomplete (Mahinpei et al., 2021). To address this, *Concept Embedding Models* (CEMs) (Espinosa Zarlenga et al., 2022) use high-dimensional embeddings $\hat{\mathbf{c}}_i \in \mathbb{R}^z$ to represent concepts alongside their truth degrees $\hat{c}_i \in [0, 1]$.

## 3 CAUSAL CONCEPT GRAPH MODELS

**Problem:** This work addresses causal opacity, the challenge of uncovering a DL model's hidden causal structure. **Research question:** To overcome this, we aim to design DL models where each prediction can be traced back to a chain of semantically meaningful causes. **Challenged assumptions:** In pursuing this goal, we challenge two key assumptions of Concept Bottleneck Models (CBMs): i) humans know a priori which interpretable variables (concepts) are direct causes of a class label (and datasets embed this prior knowledge), and ii) all interpretable variables used for explanations (i.e., concepts) are conditionally independent. **Solution:** We introduce Causal Concept Graph Models (Causal CGMs), a class of models making class predictions through cause-effect chains of interpretable variables. We first introduce Causal CGMs, then show how their decision-making process is causally transparent and conclude by describing their learning process.

### 3.1 BLUEPRINT

Dropping the assumptions of CBMs—that concepts are independent and direct causes of class labels—requires a model able to capture all possible dependencies[3] between interpretable variables

---

[2] https://github.com/gabriele-dominici/CausalCGM

[3] This naive formulation leads to a cyclic probabilistic model, making computation intractable due to the difficulty of handling cycles. Causal Concept Graph Models address this by ensuring efficient learning and producing simple explanations by design (see Sec. 3.3).

(i.e., any classification label). To model such dependencies, Concept Graph Models duplicate the layer of interpretable variables $\mathcal{V}$ by introducing an additional layer of identical copies $\mathcal{V}'$[4]. Using this additional layer, Causal CGMs make predictions of each $v_i \in \mathcal{V}$ using as possible inputs all $v'_j \in \mathcal{V}'$ for all $j \neq i$. This inference structure enables CGMs to trace each prediction in $\mathcal{V}$ back to a set of interpretable variables in $\mathcal{V}'$. As a result, the set of copies $\mathcal{V}'$ in a Causal CGM play the role of "explaining variables", akin to a CBM's concepts, while the set of variables $\mathcal{V}$ play the role of "explained variables", akin to a CBM's tasks.

**Definition 3.1** (Causal Concept Graph Model). Given an observed feature vector $\mathbf{x}$, a set of $k \in \mathbb{N}$ latent factors $u_i \in \mathcal{U}$ each associated with a pair of identical high-level interpretable variables $v_i \in \mathcal{V}$ and $v'_i \in \mathcal{V}'$ with $v_i = v'_i$, $\forall i \in \{1, \dots, k\}$, a *Causal Concept Graph Model* $\Gamma = (\mathcal{N}, \mathcal{E}, \mathcal{F}_\theta)$ with

- nodes $\mathcal{N} = \{x\} \cup \mathcal{U} \cup \mathcal{V}' \cup \mathcal{V}$,

- edges $\mathcal{E} = \big\{(\mathbf{x}, u_i) \mid u_i \in \mathcal{U}\big\} \cup \big\{(u_i, v'_i) \mid i \in \{1, \cdots, k\}\big\} \cup \big\{(u_i, v_i) \mid i \in \{1, \cdots, k\}\big\} \cup \{(v'_i, v_j) \mid v'_i \in \mathcal{V}', v_j \in \mathcal{V}, i \neq j\}$,

- and mechanisms $\mathcal{F}_\theta = \{\zeta, s, f\}$,

represents the joint conditional distribution:

$$p(v_1 \dots, v_k, v'_1, \dots, v'_k, u_1, \dots, u_k \mid x; \theta) = \prod_{i=1}^{k} \overbrace{p(v_i \mid \mathrm{pa}_{\mathcal{V}'}(v_i), u_i; \theta_f)}^{\text{endogenous predictor}} \prod_{j \neq i} \overbrace{p(v'_j \mid u_j; \theta_s)}^{\text{copies predictor}} \overbrace{p(u_j \mid \mathbf{x}; \theta_\zeta)}^{\text{exogenous encoder}}$$

$$(1)$$

## 3.2 Causal transparency in Concept Graph Models

Notice how the distribution $p(v_i \mid \mathrm{pa}_{\mathcal{V}'}(v_i), u_i; \theta_f)$ can be associated with a structural causal model $\mathcal{M}_{CGM} = (\{u_i\}, \mathcal{V}' \cup \{v_i\}, \{f_i\})$ with causal mechanism $f_i$ where $u_i \in \mathcal{U}$ is an exogenous variable representing latent, uninterpretable information (e.g., noise), $v_i \in \mathcal{V}$ is an endogenous variable representing interpretable, symbolic information, and $f_i : U \times V \to V$ is a function describing the causal mechanism that determines the value of $v_i$ given its parents $\mathrm{pa}_{\mathcal{V}'}(v_i)$. Such dependencies are captured by a graph $\mathcal{G}' = (\mathcal{V} \cup \mathcal{V}', \{(v'_i, v_j) \mid v'_i \in \mathcal{V}', v_j \in \mathcal{V}, v'_i \neq v_j\})$, representing all direct causal dependencies between endogenous variables. By modelling such dependencies, and considering that by Def. 3.1 the variables $v_i$ and $v'_i$ are identical copies of the same interpretable variable, Causal CGMs allow any endogenous variable $v_i$ to be expressed in terms of its parents in $\mathcal{V}$, denoted as $\mathrm{pa}_{\mathcal{V}}(v_i)$, rather than in terms of its parents in $\mathcal{V}'$, i.e., $\mathrm{pa}_{\mathcal{V}'}(v_i)$, as formalised in the following theorem (proof in App. C).

**Theorem 3.2.** *Given a Causal Concept Graph Model $\Gamma = (\mathcal{N}, \mathcal{E}, \mathcal{F}_\theta)$, let the set of endogenous root nodes be defined as: $roots(\mathcal{G}') = \{v'_i \in \mathcal{V}' \mid \nexists(v'_j, v_i) \in \mathcal{E}_{\mathcal{G}'}\}$, and the set of children of root nodes as: $ch(roots(\mathcal{G}')) = \{v_i \in \mathcal{V} \mid \exists(v'_j, v_i) \in \mathcal{E}_{\mathcal{G}'}, v'_j \in roots(\mathcal{G}')\}$. Then, for all nodes in the set $\mathcal{V} \setminus \{roots(\mathcal{G}') \cup ch(roots(\mathcal{G}'))\}$, i.e., for all nodes that are neither root nodes nor children of root nodes, the following holds:*

$$p(v_i \mid pa_{\mathcal{V}'}(v_i), u_i; \theta_f) = p(v_i \mid pa_{\mathcal{V}}(v_i), u_i; \theta_f)$$

The above theorem shows that CGM's bipartite graph $\mathcal{G}'$ between nodes $\mathcal{V}'$ and $\mathcal{V}$ can efficiently represent arbitrary long cause-effect chains among interpretable variables (example in Fig. 2). While during training, CGMs exploit the efficient bipartite representation, at test time CGMs can be unfolded following the graph as described in Theorem 3.2 (Sec. 3.3 describes how to learn sparse and acyclic dependencies to make the test time unfolding more efficient and interpretable; we provide more

---

[4]A detailed analysis of this transformation, including the "unfolding" of the original cyclic model, is provided in App. B and a neural parametrization in App. D.

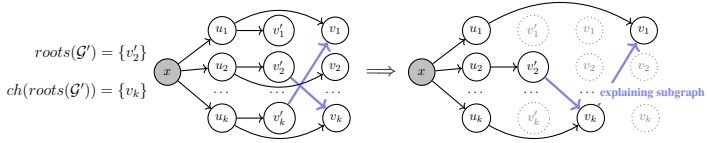

Figure 2: $p(v_1 \mid v'_k, u_1; \theta_f)$ is equivalent to $p(v_1 \mid v_k, u_1; \theta_f)$ as they both represent the same query on the same conditional probability table.

details in App. H.6). This property makes CGMs causally transparent by design, as they can make class predictions through cause-effect chains of endogenous variables, allowing humans to trace each prediction back to its underlying causes within an interpretable parental subgraph. Notice how—unlike CBMs—Causal CGMs do not require to know in advance cause (concept)-effect (task) relationships, nor require that all causes (concepts) are independent from each other, as cause-effect dependencies among interpretable variables are not given, but learnt during training.

*Remark* 3.3. Causal CGMs are designed to address the problem of causal opacity, not causal discovery. Causal opacity—the problem of understanding a model's decision-making process— neither implies nor requires causal discovery—the problem of identifying the rules governing the world or the data-generating mechanism. Indeed, the causal structure of a model's decision-making process does not necessarily mirror the causal structure of the world (Boge, 2022). We report in appendices extended discussions on CGM's relations with causal discovery (App. J) and causal representation learning (App. K), and on CGM's identifiability (App. L).

## 3.3 TRAINING CAUSAL CGMs

**Learning sparse, directed, and acyclic structures** In order to construct simple explanations, we can make Causal CGMs' graphs sparse and acyclic. To this end, we parametrise the bipartite graph $\mathcal{G}'$ with an adjacency matrix of learnable weights $M \in \mathbb{R}^{k \times k}$. We initialise the weights $m_{ij} \in M$ based on the conditional entropy between labels $v_i$ and $v_j$ to account for asymmetric concept dependencies (for other initialisation strategies, see App. F). These weights are then fine-tuned through an end-to-end learning process within the Causal CGM framework following common causal priors, where causal graphs are assumed to be sparse, directed, and acyclic (forming a Directed Acyclic Graph or DAG). We introduce a parameter $\gamma \in \mathbb{R}$ to eliminate less significant dependencies and a loss function, as described by Yang et al. (2020), to enforce the sparsity and acyclicity of the causal graph, ensuring that the adjacency matrix $A$ effectively represents a DAG:

$$\text{(initialisation)} \quad a_{ij} = -\sum_{b \in \{0,1\}} \sum_{c \in \{0,1\}} p(v_i = b, v_j = c) \log p(v_i = b \mid v_j = c) \quad (2)$$

$$\text{(sparsity)} \quad A = M \cdot \mathbb{1}_{M \geq \gamma} \quad (3)$$

$$\text{(acyclicity)} \quad \mathcal{L}_2(A) = \text{Tr}\left(\left(I + \frac{\beta}{k} A \cdot A\right)^k\right) - k \quad (4)$$

where $k$ is the number of endogenous, $\mathbb{1}$ an indicator function, and $\beta > 0$ a scaling hyperparameter.

**Optimisation problem** We can now state the general learning objective for Causal CGMs. Given (1) a set of entities represented by their feature vectors $\mathbf{x} \in \mathcal{X} \subseteq X$ (i.e. *the input*) and (2) a set of annotations for each exogenous variable $v \in \mathcal{V} \subseteq V$ (i.e. *the labels*), we wish to find functions $\zeta$, $s$, $f$, together with the adjacency matrix $A$, that maximise the log-likelihood of $v, v'$, while observing $x$ (or equivalently $u = \zeta(x)$):

$$\mathcal{L} = \overbrace{\mathbb{E}_{u,v' \sim p(u,v')}[-\log p(v' \mid u)]}^{\text{endogenous copies' prediction}} + \lambda_1 \overbrace{\mathbb{E}_{v \sim p(v \mid do(v'=v),u)}[-\log p(v \mid do(v'=v), u)]}^{\text{endogenous variables' prediction}} + \lambda_2 \overbrace{\mathcal{L}_2(A)}^{\text{graph priors}}$$

where $\lambda_{1,2}$ are hyperparameters balancing optimisation objectives (see related ablation studies in App. H.3). Notice that we use the *do*-operator replacing $\hat{v}'_j$ with labels $v_j$ to reduce leakage and provide better gradients to the endogenous predictor $f$. This enables Causal CGM to be aware of *do*-operations during training, making it effective in responding to *do*-interventions once deployed.

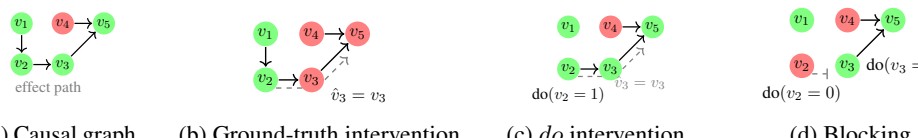

(a) Causal graph.  (b) Ground-truth intervention.  (c) *do* intervention.  (d) Blocking.

Figure 3: (a) A 5-variable causal graph. (b) A ground-truth intervention fixes the error of the prediction $\hat{v}_3$ to the ground-truth label $v_3$. (c) A do-intervention sets the value of the second variable to a constant i.e., $v_2 = 0$. The intervention impacts $v_2$'s *effects* i.e., $v_{3,5}$, but does not alter $v_2$'s *causes* i.e., $v_1$. This operation can override ground-truth interventions. (d) A do-intervention on $v_3$ *blocks* the causal effects of $v_2$ on $v_5$. As a result, intervening on $v_2$ cannot alter $v_5$ anymore.

### 3.4 HIGH-DIMENSIONAL ENDOGENOUS REPRESENTATION

As observed by Mahinpei et al. (2021), using scalar representations for concepts (corresponding to endogenous copies in our context) can significantly degrade predictive performance in realistic settings. To address this issue, and inspired by Espinosa Zarlenga et al. (2022), Causal CGMs employ high-dimensional representations for endogenous copies $\mathbf{v}'_j$. For each endogenous copy, Causal CGMs learn a mixture of two embeddings with explicit semantics, representing the variable's state. This design enables the model to construct evidence both in favour of and against a variable's state and supports simple concept interventions, as it allows switching between the two embedding states during interventions. Specifically, we represent the context of each variable with two embeddings: $[\mathbf{c}^+_j, \mathbf{c}^-_j] = \mathbf{u}_j = \zeta(\mathbf{x}), \quad \mathbf{c}^+_j, \mathbf{c}^-_j \in \mathbb{R}^m$. Each embedding carries specific semantics: $\mathbf{c}^+_j$ represents the active state of the variable, while $\mathbf{c}^-_j$ represents the inactive state. Once these semantic embeddings are computed, the final endogenous embedding $\mathbf{v}'_j$ for $v_j$ is constructed as a mixture of $\mathbf{c}^+_j$ and $\mathbf{c}^-_j$, weighted by the endogenous state:

$$\mathbf{v}'_j = v'_j \mathbf{c}^+_j + (1 - v'_j)\mathbf{c}^-_j \tag{5}$$

This formulation serves two primary purposes: (i) it forces the model to rely exclusively on $\mathbf{c}^+_j$ when the $j$-th endogenous variable is active ($v'_j = 1$) and on $\mathbf{c}^-_j$ when inactive, thereby creating two distinct and semantically meaningful latent spaces; (ii) it enables a straightforward intervention strategy, where one can switch between embedding states to correct a mispredicted endogenous variable.

### 3.5 CAUSAL REASONING AND VERIFICATION WITH CAUSAL CGM

CGMs support two kinds of interventions: ground-truth interventions, as in CBMs, and do-interventions, as in causal models. In what follows, we first highlight how CGM's design makes ground-truth interventions more effective and then we describe how do-interventions can be applied in Causal CGMs to verify properties of the model's decision-making process.

**Ground-truth interventions** Causal CGMs support "ground-truth interventions" (see Figure 3b). Ground-truth interventions are one of the core motivations behind CBMs (Koh et al., 2020). Through ground-truth interventions, concept bottleneck models allow experts to improve a CBM's task performance by rectifying mispredicted concepts at test time, thus significantly improving task performance within a human-in-the-loop setting. In Causal CGMs, however, ground-truth interventions have a potential impact on all endogenous variables descendant of an intervened node, which may include not only nodes corresponding to downstream tasks but also nodes corresponding to a CBM's intermediate concepts. This enables a single concept ground-truth intervention to potentially improve the prediction of intermediate concepts as well as downstream tasks.

**Causal reasoning and verification: do-interventions, counterfactuals, and blocking** Causal CGMs can answer interventional and counterfactual queries related to the model's decision-making process using the do-operator on the unfolded SCM. **Do-interventions** enable manipulation of a Causal CGM's decision-making process by changing the value of a specific endogenous variable and observing how it affects other variables' distributions (see Figure 3c). In Causal CGMs, the effect of the *do*-intervention is analysed through the interventional distribution, denoted as $p(v_i|do(v_j = \kappa), \text{pa}(v_i))$, which describes the distribution of the outcome variable $v_i$ after the intervention $do(v_j = \kappa)$ has been performed. In particular, in Causal CGMs, the do-operation fixes the value of the

intervened variable $v_i$ to a fixed constant $\kappa \in \{0, 1\}$ and removes all causal dependencies from parent variables by zeroing all values of the $i$-th column of the adjacency matrix:

$$\text{do}(v_j = \kappa) := \begin{cases} v_j := \kappa, & \kappa \in \{0, 1\} \\ a_{[:,j]} = 0, & (\text{implies that: } \text{pa}(v_j) = \emptyset) \end{cases} \tag{6}$$

Causal CGMs also enable to answer **counterfactual** queries such as "What would the value of the $i$-th variable have been, had the $j$-th variable been $\kappa$, given that we observed $v_i$ and $v_j$?" (extended motivation in App I). Answering these queries involves three steps (Pearl, 1995): 1) *Abduction*: Infer a realisation of exogenous variables that is consistent with the observed $v_i$ and $v_j$ in the actual causal model. 2) *Action*: Modify the architecture of the Causal CGM $\mathcal{M}$ into $\mathcal{M}_{v_j = \kappa}$ by replacing the structural equation for $v_j$ with $\kappa$, to simulate the intervention. 3) *Prediction*: Compute the value of $v_i$ in the modified model $\mathcal{M}_\kappa$, representing the counterfactual outcome:

$$(abduction) \quad \hat{u}_j = p(u_j \mid \mathbf{x}; \theta_\zeta) \tag{7}$$

$$(action) \quad \text{do}(v_j = \kappa) := \begin{cases} v_j := \kappa \\ a_{[:,j]} = 0 \end{cases} \tag{8}$$

$$(prediction) \quad \hat{v}_i = p(\hat{v}_i \mid \text{pa}_\mathcal{V}(v_i), u_i; \theta_f) \; \forall i \neq j \tag{9}$$

This formalism allows us to not only estimate the effects of hypothetical interventions but also to explore the implications of alternative scenarios on individual outcomes, providing a powerful tool for analysing the model's decision-making based on interpretable causal structures.

**Model verification and blocking** Causal analysis enables the verification of properties of Causal CGMs before deployment. For instance, using only the learnt causal graph, it is possible to prove that an endogenous variable $v_i$ is independent of the variable $v_j$ by verifying that $v_j$ is not among the ancestors of $v_i$. Another form of formal verification, which we call "blocking", employs the $do$-intervention (see Figure 3d). Blocking allows one to formally verify the independence of a pair of variables given a sequence of $do$-operations. Given a pair of variables $v_j$ and $v_i$ such that $v_j$ is an ancestor of $v_i$, we perform a blocking verification as follows: 1) *Block*: perform a do-intervention on all child nodes of $v_j$, 2) *Verify*: perform a do-operation on $v_j$ itself and observe the impact on $v_i$. We can easily verify that the first step makes $v_j$ and $v_i$ completely independent by observing that the do-operation on $v_j$ no longer alters the distribution of $v_j$.

## 4 EXPERIMENTS

Our experiments aim to answer the following questions:

- **Concept-based performance and interpretability:** Can Causal CGMs match the generalisation performance of equivalent black-box models and existing CBMs? Can Causal CGMs enable more effective ground-truth interventions w.r.t. existing CBMs?

- **Causal Interpretability:** Are Causal CGMs causally interpretable? Can Causal CGMs effectively block the causal effect of two causally related endogenous variables?

To answer these questions, we use four datasets: (i) Checkmark, a synthetic dataset composed of four endogenous variables; (ii) dSprites, where endogenous variables correspond to object types together with their position, colour, and shape; (iii) CelebA, a facial recognition dataset where endogenous variables represent facial attributes; (iv) CIFAR10, an animal classification dataset where the endogenous variables are extracted automatically following Oikarinen et al. (2023). Using these datasets, we compare the proposed approach with a **black box** baseline and state-of-the-art concept-based architectures: Concept Bottleneck Models (**CBM**) (Koh et al., 2020), and Concept Embedding Models (**CEM**) (Espinosa Zarlenga et al., 2022). We also compare a version of the proposed method (**Causal CGM**) where the causal graph is learnt end-to-end w.r.t. with a version (**Causal CGM+CD**) where the causal graph is extracted from ground-truth labels using a causal discovery algorithm (Lam et al., 2022) and injected into the model fixing the matrix $A$. Due to the prohibitive computational complexity of the causal discovery method, this was not feasible for the CIFAR-10 dataset We provide further details on our experimental setup and baselines in App. G.

A comprehensive set of experiments is detailed in App. H, where the experiments presented in this section for a subset of the datasets are extended to include all datasets. We also report an analysis of Causal CGMs' scalability and computational complexity in App. H.5.

We remark that, following the CBM literature, **the goal of Causal CGMs is not to increase performance but interpretability**. Therefore, the experiments aim to show that our Causal CGMs' classification performance is competitive against existing black-box models and state-of-the-art concept-based approaches (e.g., within one percentage point) while enabling better visibility and manipulation of its reasoning process.

## 4.1 KEY FINDINGS

**Causal CGMs match the performance of causally opaque models. (Table 1)** Causal CGMs demonstrate robust generalisation across all datasets, yielding a predictive performance close to that of black-box architectures with an equivalent capacity. Causal CGMs using a pre-trained causal graph (Causal CGM+CD) tend to have slightly better label accuracy with respect to Causal CGMs where the causal graph is learned end-to-end (Causal CGM). Causal CGMs's low variance suggests a consistent robustness with respect to weight initialisations over multiple training runs. Thanks to concept embeddings, concept incompleteness settings do not have a strong impact on Causal CGMs' performance (results in App. H.4).

Table 1: Label accuracy ($\uparrow$) is computed on all endogenous variables (concepts and task).

|  | CHECKMARK | DSPRITES | CELEBA | CIFAR10 | SEMANTIC TRANSPARENCY | CAUSAL TRANSPARENCY |
|---|---|---|---|---|---|---|
| Black box | $90.15_{\pm 1.30}$ | $99.53_{\pm 0.05}$ | $79.55_{\pm 0.14}$ | $94.85_{\pm 0.03}$ | ✗ | ✗ |
| CBM | $90.34_{\pm 0.55}$ | $99.55_{\pm 0.07}$ | $79.00_{\pm 0.18}$ | $92.17_{\pm 0.11}$ | ✓ | ✗ |
| CEM | $89.09_{\pm 1.98}$ | $99.48_{\pm 0.07}$ | $79.17_{\pm 0.26}$ | $92.04_{\pm 0.06}$ | ✓ | ✗ |
| **Causal CGM+CD** | $89.43_{\pm 0.93}$ | $99.40_{\pm 0.15}$ | $78.42_{\pm 0.42}$ | N/A | ✓ | ✓ |
| **Causal CGM** | $88.24_{\pm 1.30}$ | $99.44_{\pm 0.11}$ | $78.23_{\pm 0.45}$ | $93.32_{\pm 0.03}$ | ✓ | ✓ |

**Ground-truth interventions on Causal CGMs improve both concept and task accuracy as opposed to CBMs (Figures 4, 5)** In Causal CGM, the causal graph induces a natural strategy for ground-truth interventions. Indeed, the causal graph narrows down the set of variables to intervene upon: for any given node, we can just fix mispredicted labels of the node's ancestors as intervening on other nodes will not have any impact. This property significantly decreases the required number of interventions to achieve a desired outcome (e.g., to increase a downstream task accuracy), as shown in App. H. Another advantage of Causal CGM consists in the hierarchical nature of inference that allows ground-truth interventions to impact all endogenous variables descendant of an intervened node. In particular, ground-truth interventions may affect not only nodes corresponding to downstream tasks (as in CBM and CEM), but also nodes corresponding to a CBM's intermediate concepts. We experimentally verify this property and its impact by calculating—for nodes that were not intervened upon (including both concepts and tasks)—the change in accuracy before and after ground-truth interventions were applied on their ancestors. Our results (Figure 4 and App. H.1) show that Causal CGM improves nodes accuracy by $\sim 15$ percentage points after only 7 ground-truth interventions on CelebA. CBM and CEM, instead, achieve a similar performance only after intervening on all

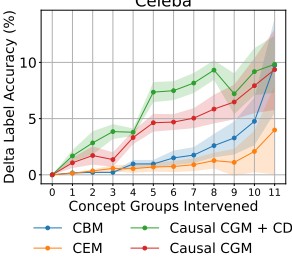

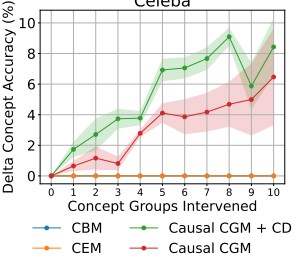

Figure 4: Impact of ground-truth interventions on non-intervened nodes ($\uparrow$). Intervention on Causal CGM improves both concept and task accuracy.

Figure 5: Impact of ground-truth interventions on concept nodes ($\uparrow$). Intervention on CBM and CEM do not improve concept accuracy unlike Causal CGM.

concepts as their architecture assumes all concepts to be mutually independent. Focusing solely on concept-level analysis, Causal CGM enhances concept accuracy by up to $\sim 25$ percentage points, compared to CBM and CEM, where a concept intervention does not impact the accuracy of other concepts (Figure 5). Causal CGM's advantage increases with the number of concepts and connections, as a single intervention can impact a higher number of nodes in the causal graph.

**Causal CGMs' endogenous predictors are causally interpretable (Figures 6a, 6b)** In Causal CGM the decision-making process is causally interpretable and can be analysed by visualising the learnt causal graph and structural equations as in a structural causal model, as shown in Figure 6a for CelebA and in Figure 6b for Cifar10 (more results in App. H.2). The first image shows how Causal CGM exploited known biases in CelebA to infer facial attributes. For instance, Causal CGM predicts the attribute "wearing lipstick" directly from the attribute "attractive" and indirectly from attributes such as "smiling" or "high cheek", all attributes that are known to be strongly correlated with each other in CelebA (Ramaswamy et al., 2021; Wang & Russakovsky, 2023). Similarly, in the CIFAR-10 dataset, where concepts are automatically extracted, the model learns meaningful relationships between concepts (Figure 6b), such as the presence of a "port" implying the presence of a "dock", or the connection between a "beak" and a "bird". We can also quantify the strength of the causal dependency between two nodes by computing the probability of necessity and sufficiency (PNS) (Pearl, 2022). In the figure, we represent the PNS w.r.t. the leaf node by colouring each node with a different shade of orange (more PNS of the other datasets in App. I). This shows how, for Causal CGM, the attribute "heavy makeup" has the strongest impact on the leaf node. This high degree of causal transparency allows users to interpret Causal CGM's inference and can be eventually exploited to identify potential biases, thereby supporting the assessment of the model's counterfactual fairness (an example of generated counterfactuals in Table 2). As a result, users can intervene directly on the causal structure of the decision-making process and remove biases using do-interventions, as shown in the next paragraph.

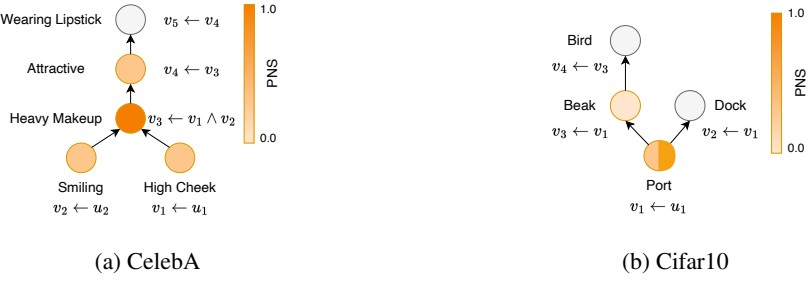

|  (a) CelebA  |  (b) Cifar10  |

Figure 6: Portion of the learnt causal graph and structural equations in CelebA (on the left) and Cifar10 (on the right). A node's colour in the causal graph is proportional to the probability of necessity and sufficiency w.r.t. to the child nodes.

Table 2: Examples of counterfactuals generated on CelebA obtained via do-interventions (intervened variables are marked in red, changed variables are underlined).

|  | Endogenous variables' activations |
|---|---|
| Initial Predicted State | Smiling, Not Attractive, Mouth Slig, High Cheek, Not Wearing Li, Not Heavy Make, Not Wavy Hair, Not Big Lips, Not Oval Face, Not Makeup, Not Fem Model |
| What if I would wear lipstick? | Smiling, Attractive, Mouth Slig, High Cheek, Wearing Li, Heavy Make, Not Wavy Hair, Not Big Lips, Not Oval Face, Not Makeup, Not Fem Model |
| What if I would wear lipstick and also makeup? | Smiling, Attractive, Mouth Slig, High Cheek, Wearing Li, Heavy Make, Not Wavy Hair, Big Lips, Not Oval Face, Makeup, Fem Model |

**Causal CGMs can make two causally-related variables causally independent by blocking all paths between these variables (Table 3)** The causal transparency of the proposed approach allows users to modify the model's decision-making process (e.g., to de-bias the model's inference) by using do-interventions. In particular, we can make two causally-related variables $i$ and $j$ causally independent by blocking all paths between the cause $i$ and the effect $j$. We experimentally verify this property and its impact by computing the Residual Concept Causal Effect i.e., the ratio between the Concept Causal Effect (CaCE) (Goyal et al., 2020) obtained after and before blocking. The optimal value of this metric is zero, corresponding to perfect causal independence between $i$ and $j$ (i.e., the

optimal value for a de-biasing operation). The experimental procedure is reported in App. H.7. The results show that, in Causal CGMs, blocking a variable in the causal graph always yields a perfect Residual Concept Causal Effect of zero across all datasets. In contrast, applying the same procedure in CBMs leads only to a negligible reduction in the average causal effect to 3 percentage points. CEMs not only fail to reduce the causal effect to zero but, in some cases, even increase the causal influence. These results underscore how Causal CGMs transparency enable users to manipulate the model's decision-making process to achieve desired outcomes, as opposed to existing CBMs.

Table 3: Residual Concept Causal Effect ($\downarrow$) between causally-related variables having blocked all paths between the two variables with do-interventions on the causal graph. The optimal value is zero corresponding to perfect causal independence. (see App. H.7 for further clarification)Values above $100\%$ mean that the causal effect increased instead of decreasing.

|  | CHECKMARK | DSPRITES | CELEBA | CIFAR10 |
|---|---|---|---|---|
| Black box | N/A | N/A | N/A | N/A |
| CBM | $97.99_{\pm 5.64}$ | $100.00_{\pm 0.70}$ | $97.84_{\pm 2.13}$ | $100.00_{\pm 0.00}$ |
| CEM | $102.58_{\pm 12.95}$ | $100.00_{\pm 4.62}$ | $106.00_{\pm 0.50}$ | $100.00_{\pm 0.00}$ |
| **Causal CGM+CD** | $0.00_{\pm 0.00}$ | $0.00_{\pm 0.00}$ | $0.00_{\pm 0.00}$ | N/A |
| **Causal CGM** | $0.00_{\pm 0.00}$ | $0.00_{\pm 0.00}$ | $0.00_{\pm 0.00}$ | $0.00_{\pm 0.00}$ |

## 5 DISCUSSION

**Related works** Causal CGMs present substantial advantages compared to the state of the art. Compared to most causal feature-attribution methods (e.g., (Chattopadhyay et al., 2019)), Causal CGMs focus on high-level human-interpretable concepts. Causal CGMs differ from existing CBMs in their approach to intervention and causal relationships. Unlike vanilla CBMs, where concepts are direct, independent causes of the target, Causal CGMs incorporate richer causal structures. This enables human-in-the-loop corrections to mispredicted reasoning steps, improving both downstream accuracy and explanation fidelity. Causal CGMs are also more flexible than existing relational CBMs (Barbiero et al., 2024) where concept relations are assumed to be provided a priori. The closest methods to Causal CGMs are *post-hoc* causal concept-based explainability techniques, like DiConStruct (de Oliveira Moreira et al., 2024) and conceptual counterfactual explanations (Abid et al., 2021). These use surrogate causal models to emulate a black box's predictions. However, as Rudin (2019) notes, matching predictive behaviour does not guarantee structural similarity in decision-making, making surrogate models unreliable explanatory proxies. Causally interpretable *by-design* architectures, such as Causal CGMs, do not suffer from this issue.

**Limitations and future works** Our method's limitations stem from those of CBMs and causal reasoning. The quality of learnt causal graphs depends on dataset quality and annotations; missing or noisy labels may lead to suboptimal graphs. Like generalised PGMs, Causal CGMs unfold easily when the final graph is acyclic. Cyclic variables remain inferable but require special unfolding techniques, which future work may explore. Finally, a Causal CGM's learnt graph represents the model's inference, not necessarily the data-generating process, making it suitable for verification and control but not for understanding the dataset's distribution.

## 6 CONCLUSION

Causal opacity represents a key open challenge at the intersection of deep learning, interpretability, and causality. Causal CGMs address this challenge by employing an architecture which makes the decision-making process causally transparent by design. This makes Causal CGMs reliable and verifiable compared to both usual DL architectures and standard (non-causal) CBMs. The results of our experiments show that Causal CGMs support the analysis of interventional and counterfactual scenarios—thereby improving the model's causal interpretability and supporting the effective verification of its reliability and fairness—and enable human-in-the-loop corrections to mispredicted intermediate reasoning steps, boosting not just downstream accuracy after corrections, but also accuracy of the explanation provided for a specific instance. As a result, advancing this research line could significantly improve the reliability and verifiability of concept-based deep learning models, thus supporting their deployment in real-world applications.

ACKNOWLEDGMENTS

GD acknowledges support from the European Union's Horizon Europe project SmartCHANGE (No. 101080965). PB acknowledges support from Swiss National Science Foundation projects TRUST-ME (No. 205121L_214991) and IMAGINE (No. TMPFP2_224226). MEZ acknowledges support from the Gates Cambridge Trust via a Gates Cambridge Scholarship. AT acknowledges the support by the Hasler Foundation grant Malescamo (No. 22050), and the Horizon Europe grant Automotif (No. 101147693). MG acknowledges support from Swiss National Science Foundation projects XAI-PAC (No. PZ00P2_216405). GM acknowledges support from the KU Leuven Research Fund (STG/22/021, CELSA/24/008) and from the Flemish Government under the "Onderzoeksprogramma Artificiële Intelligentie (AI) Vlaanderen" programme. This work was also supported by the EU Framework Program for Research and Innovation Horizon under the Grant Agreement No 101073307 (MSCA-DN LeMuR).

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

## A    CAUSAL EXPLAINABILITY, CAUSAL OPACITY, AND CAUSAL DISCOVERY

Deep Learning (DL) models have a pervasive impact on many areas of contemporary research and society (Baldi, 2021). Despite this success, there is growing concern about the widespread real-world application of DL, particularly in sensitive domains (Dignum, 2019). This is partly due to lack of *causal explainability* of these models, which undermine their robustness, fairness and generalisability in out-of-distribution context (Schölkopf et al., 2021). Causal explainability, in particular, is a multi-faced issue, which includes a variety of different, albeit related, problems (Termine & Primiero, 2024). One such problem is that of *causal discovery* and concerns the possibility of using a model to detect and understand the *causal mechanisms* of the data generating process (Pearl & Mackenzie, 2018; Pearl, 2019; Schölkopf et al., 2021; Kaddour et al., 2022). An equally important but **distinct** problem is that of *causal opacity*, which denotes the difficulty of users to grasp and understand the "hidden" *causal structure* that underlines the predictions delivered by a given model (see Figure 1a). This problem can be better understood in the light of Pearl's framework of causality (Pearl, 2009; 2019), which relates causal understanding to the ability of an agent to answer *what-if* type of questions. In particular, Pearl identifies three different kinds of what-if questions organised in a hierarchy of levels. At the bottom level we have *observational questions*, which concern the actual state of affairs one can observe in data. At the intermediate level we find *interventional questions*, which concern the effects of intervening on the actual state of affairs and fix the value of some variable to a pre-defined one. Finally, the top level encompasses *counterfactual questions*, which concern an hypothetical state of affairs that could have been occured but it is not the actual one. The three levels provide a measure of the causal understanding that an agent has of a given target-system. Agents that answer only observational questions do not possess any causal understanding of the target system; agents with an intermediate level of causal understanding can answer interventional questions, whereas a complete causal understanding is required for counterfactual questions.

In this paper, we propose to involve Pearl's hierarchy to define and measure the causal opacity/transparency of a DNN model based on the ability of its users to answer interventional and counterfactual questions concerning the structure of its inferential behaviour (e.g., "what happens if I fix the feature *age* to a value greater than $50$?" or "what the model's prediction would have been if the feature *age* had taken a value greater than $50$ instead of lower then or equals to $50$?"). In this regard, complete causal transparency require that users can answer both interventional and counterfactual questions concerning the model's inferential behaviour, whereas if only observational questions can be addressed (i.e., questions regarding the input-output associations of the model), then the model results completely causally opaque: this is notably the case of the majority of DNNs considered in contemporary AI research.

Notice that causal opacity is fundamentally **distinct** from the problem of causal discovery, which concerns the causal structure *of the world* and not that of a model's inference. To clarify this point, consider a simple rules-based model as the one described in Fig. 7.

| Risk_hearth_attack( = high) ← Age( ≥ 50) ∨ Income( ≤ 100.000) |
| Risk_hearth_attack( = low) ← Age( < 50) ∨ Income( > 100.000) |

Figure 7: Rules-based Model $M_1$

This model is causally transparent as users can easily compute interventional and counterfactual questions concerning its inferential behaviour (e.g., "what if the age of the patient would had been $\geq 50$ instead of $< 50$"?). However, causal transparency does not imply that the causal structure of the model's inference (Fig.8a) resemble the causal structure of the world (Fig. 8b), which remains a very distinct animal.

To ensure that a model is causally transparent and matches the causal structure of the world, indeed, a suitable combination of causal transparency and causal discovery techniques is usually required. However, causal transparency *per-sé* is a fundamental requirement and sufficient for a variety of tasks, such as establishing the model's fairness (especially within the popular framework of *counterfactual fairness* (Kusner et al., 2017)). Consider the model in Fig. 9, which is a slightly modified version of the model depicted in Fig. 7. The causal transparency of the model makes it easy for users to verify the un-fairness of its inferential behaviour (stemming from the fact that the model uses the variable

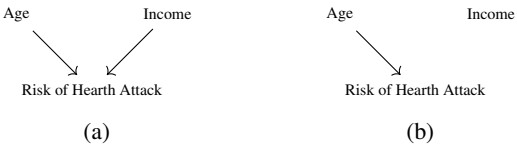

Figure 8: A graphical representation of the causal structure of $M_1$'s inferential behaviour (a) as opposed to the real causal structure of the world (b). As $M_1$ uses the variable *Income* as a predictor of the variable *Risk of Hearth Attack*, a causal edge is present between the two variables in *a*. This edge is not present in *b* as no direct causal connection exists between the two variables in the real world.

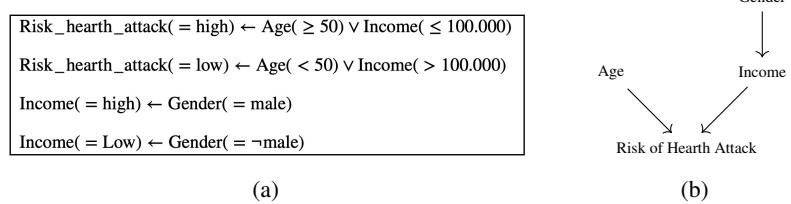

Figure 9: (a) Slightly modified version of $M_1$, and (b) a graphical representation of the causal structure of its inferential behaviour.

*gender* to predict the variable *income*), and implement suitable corrections (e.g. remove the causal link between gender and income).

## B    DERIVATION OF CAUSAL CGMS

In the absence of causal priors, Causal CGMs need to learn causal dependencies between high-level features that lead to the model becoming more effective in solving its designated task. As a result, we can formally describe Causal CGMs using a generalised probabilistic graphical model (PGM) that extends a traditional PGM by allowing cycles:

**Definition B.1** (Generalised Causal Concept Graph Model). Given an observed input feature $x$, a set of $k \in \mathbb{N}$ latent factors $u_i \in \mathcal{U}$ each associated with a high level interpretable variable $v_i \in \mathcal{V}$, a *Generalised Causal Concept Graph Model* is the generalised probabilistic graphical model (PGM) $G = (\mathcal{N}, \mathcal{E})$ with nodes $\mathcal{N} = \{x\} \cup \mathcal{U} \cup \mathcal{V}$ and edges $\mathcal{E} = \big\{(x, u_i) \mid u_i \in \mathcal{U}\big\} \cup \big\{(u_i, v_i) \mid i \in \{1, \cdots, k\}\big\} \cup \big\{(v_i, v_j) \mid v_i, v_j \in \mathcal{V}, v_i \neq v_j\big\}$ which represents the joint conditional distribution $p(v, u \mid x)$:

$$\tag{10}$$

The cyclical nature of Causal CGMs comes from the necessity to model all possible dependencies among variables $v_i$. Notice that the model is uniquely identified by the set of all conditional probability distributions corresponding to the arrows in the graph. Unfortunately, in generalised PGMs, the model does not easily factorise in terms of such distributions due to the cycles. To deal with cycles while maintaining the independencies induced by the graph structure, we can use an unfolding semantics for cyclical PGMs (Baier et al., 2022). This semantics is based on the choice of a "cutset" i.e., a specific set of nodes $\mathcal{Q} \subseteq \mathcal{N}$ in the PGM such that every cycle in the PGM contains at least one node in $\mathcal{Q}$. Intuitively, by unfolding the nodes in the cutset, all cycles are broken leaving us with a standard acyclic PGM. The consistency between the semantics of the original cyclical PGM and the unfolded acyclic PGM is only valid in the limit of infinite unfolding (Baier et al., 2022). However, when computing the likelihood of an observed complete set of variables $v_i \in \mathcal{V}$, modelling one single unfolding (i.e., a single transition) suffices for learning the conditional probability distributions

among the variables in $\mathcal{V}$, as all the variables become conditionally independent on each other. As a result, we can define a *Dissected Causal CGM* as the one-step unfolding of the Causal CGM in Definition B.1:

**Definition B.2** (Dissected Causal CGM). Given a Causal CGM, let $\mathcal{V}' = \mathcal{V}$ be the cutset. Then, the dissected Causal CGM $\mathcal{G} = (\mathcal{N} \cup \mathcal{V}', \mathcal{E}_{\mathcal{V}'})$ is an acyclic PGM obtained by extending the generalised PGM by (i) adding a copy of all cutset nodes $\mathcal{V}' = \{v_i \mid v_i \in \mathcal{V}\}$, (ii) adding a new set of edges directed from parents of cutset's nodes to the generated copies $\mathcal{V}'$ i.e., $\mathcal{E}_{\mathcal{V}'} = \{(a, b) \mid (a, b) \in \mathcal{E}, b \in \mathcal{V}'\} \cup \{(a, b) \mid (a, b) \in \mathcal{E}, b \notin \mathcal{V}'\}$, and (iii) defining an initial probability distribution for the new copies $p(v'|u)$ given the latent variables. The resulting PGM, factorised as $p(v, v', u \mid x) = p(v \mid v', u)p(v' \mid u)p(u \mid x)$, is:

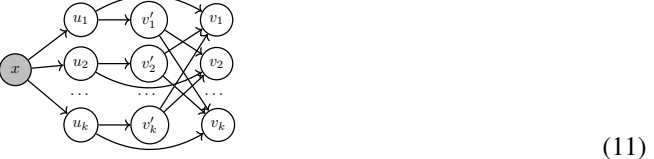

$$(11)$$

## C  PROOF OF THEOREM 3.2

*Proof.* Let us consider the graph $\mathcal{G}^* = (\mathcal{V} \cup roots(\mathcal{G}'), \{(v_i, v_j) \mid v_i, v_j \in \mathcal{V}, (v_i, v_j) \in \mathcal{E}_{\mathcal{G}'}\} \cup (v_i', v_j)|v_i' \in roots(\mathcal{G}'))$. We will first prove that this graph is isomorphic to $\mathcal{G}'$ for all nodes $\mathcal{V}$ and then that the conditional probabilities are equivalent.

The graph $\mathcal{G}'$ is defined with nodes $\mathcal{V} \cup \mathcal{V}'$, where $\mathcal{V}$ represents the endogenous variables and $\mathcal{V}'$ represents copies of the same high-level variables. Hence, the set of nodes $\mathcal{V}$ is the same for both graphs by definition. In $\mathcal{G}'$, the edge set $\mathcal{E}_{\mathcal{G}'}$ consists of edges $(v_i', v_j)$ between nodes in $\mathcal{V}'$ and $\mathcal{V}$. Specifically, each $v_i' \in \mathcal{V}'$ is connected to $v_j \in \mathcal{V}$, where $v_i' \neq v_j$. In $\mathcal{G}^*$, the edge set $\mathcal{E}_{\mathcal{G}^*}$ consists of:

- edges $(v_i, v_j)$ between nodes in $\mathcal{V}$, which are present if $(v_i, v_j) \in \mathcal{E}_{\mathcal{G}'}$,

- edges $(v_i', v_j)$ where $v_i' \in \text{roots}(G')$, representing connections between the root nodes and their corresponding child nodes.

which represent a one-to-one correspondence between the edges in $\mathcal{E}_{\mathcal{G}'}$ and $\mathcal{E}_{\mathcal{G}^*}$. Thus, the two graphs are isomorphic.

Now, we prove that for all nodes $v_i \in \mathcal{V} \setminus (\text{roots}(\mathcal{G}') \cup \text{ch}(\text{roots}(\mathcal{G}')))$, the conditional probabilities are equivalent: $p(v_i \mid \text{pa}_{\mathcal{V}'}(v_i), u_i; \theta_f) = p(v_i \mid \text{pa}_{\mathcal{V}}(v_i), u_i; \theta_f)$. In $\mathcal{G}'$, the conditional distribution of a node $v_i \in \mathcal{V}$ depends on its parents $\text{pa}_{\mathcal{V}'}(v_i)$, which are nodes in $\mathcal{V}'$, as well as the latent factors $u_i$. Since $\mathcal{V} = \mathcal{V}'$ by Def. 3.1 and the graphs $\mathcal{G}'$ and $\mathcal{G}^*$ are isomorphic for all $\mathcal{V}$ nodes, the parent set $\text{pa}_{\mathcal{V}'}(v_i)$ in $\mathcal{V}'$ corresponds directly to the parent set $\text{pa}_{\mathcal{V}}(v_i)$ in $\mathcal{V}$, up to the bijection between the two sets. Under full observability (the training conditions of CGMs), conditional queries of the form $p(v|\text{ all but } v)$ are just inspections of the conditional probability table. As a result, $p(v_i \mid \text{pa}_{\mathcal{V}'}(v_i), u_i; \theta_f)$ and $p(v_i \mid \text{pa}_{\mathcal{V}}(v_i), u_i; \theta_f)$ correspond to repeating the same query on the same conditional probability table. Therefore, for all nodes $v_i \in \mathcal{V} \setminus (\text{roots}(\mathcal{G}') \cup \text{ch}(\text{roots}(\mathcal{G}')))$, we can write: $p(v_i \mid \text{pa}_{\mathcal{V}'}(v_i), u_i; \theta_f) = p(v_i \mid \text{pa}_{\mathcal{V}}(v_i), u_i; \theta_f)$. □

## D  NEURAL PARAMETRIZATION OF CONCEPT GRAPH MODELS

We can interpret the factors of a dissected Causal CGM as follows: $p(u_j \mid \mathbf{x}; \theta_\zeta)$ is the **exogenous encoder**, i.e., a deterministic distribution that is parametrised by a neural network $\zeta : X \to U$. In CBMs, this function represents the input encoder. In causal CGMs we use the exogenous encoder to predict the value of exogenous variables as we consider that the observed input (e.g., an image's pixels) holds most of the contextual variability required to infer exogenous variables. For instance, an image provides information about background, lighting conditions, and object locations. All this information is used to anchor the endogenous predictor to a specific context and to correctly infer the values of the exogenous variables and the endogenous roots of the causal graph. The exogenous

encoder $\zeta$ generates exogenous variables $u_i \in \mathcal{U}$ mapping raw input features $\mathbf{x}$ (e.g., an image's pixels) to latent embeddings $\hat{\mathbf{u}}_i \in \mathbb{R}^q$, $q \in \mathbb{N}$. In practice, this process mirrors the generation of context vectors in Concept Embedding Models to avoid information bottlenecks which may negatively affect the model's accuracy (Espinosa Zarlenga et al., 2022). First, the encoder $\psi : X \to H$ maps raw features to a latent code $\mathbf{h} \in H$. Then, a pair of neural networks $\{\phi_i^+, \phi_i^-\}$ map the latent code into two different embeddings whose concatenation $[\phi_i^+(\mathbf{h}), \phi_i^-(\mathbf{h})]^T$ corresponds to the exogenous variable $\mathcal{U}_i$ of the $i$-th concept:

$$\text{(exogenous variables)} \quad \hat{\mathbf{u}}_i = \zeta(\mathbf{x}) = [\phi_i^+(\psi(\mathbf{x})), \phi_i^-(\psi(\mathbf{x}))]^T. \tag{12}$$

In contrast, $p(v_j' \mid u_j; \theta_s)$ is the **copies predictor**. This is the product of $k$ independent Bernoulli distributions whose logits are parameterised by a neural network $s : U \to V$. In CBMs, the composition of the exogenous encoder and the concept predictor is often called *concept encoder* $g = \zeta \circ s$. In causal methods, this function represents a (supervised) *causal feature learner* (Kaddour et al., 2022). The copies predictor $s$ generates endogenous copies $v_i' \in \mathcal{V}$ from latent embeddings $\hat{\mathbf{u}}_i$. This is obtained using a neural network classifier $s : U \to V$ as a scoring function as in (Espinosa Zarlenga et al., 2022):

$$\text{(endogenous copies)} \quad \hat{v}_i' = s(\hat{\mathbf{u}}_i) = \sigma(W_s \hat{\mathbf{u}}_i + \mathbf{b}_s) \tag{13}$$

Finally, $p(v_i \mid \mathrm{pa}_{\mathcal{V}'}(v_i), u_i; \theta_f)$ is the **endogenous predictor**. This distribution is the product of $k$ independent Bernoulli distributions whose logits are parameterised by a neural network $f : V^k \times U \to V$. The input to this function $\mathrm{pa}_{\mathcal{V}'}(v_i)$ (representing direct causal dependencies) is weighted by a learnable adjacency matrix $M \subseteq \mathbb{R}^{k \times k}$, where each learnable weight $a_{ij}$ models the strength of the dependency of $v_i$ from its parents $v_j'$. In CBMs this function is called a *task predictor* (Koh et al., 2020) and intuitively represents the analogous of the structural equations that model *causal mechanisms* in SCMs (Kaddour et al., 2022). The endogenous predictor generates the endogenous variables $\hat{v}_i \in \mathcal{V}$ by considering exogenous variables $\hat{\mathbf{u}}_j$ and copies $\hat{v}_j'$ with $j \neq i$. First, the function $\omega : V \times U \to \mathbb{R}^z$ generates endogenous embeddings $\hat{\mathbf{v}}_j'$ using exogenous variables $\hat{\mathbf{u}}_j$ and copies $\hat{v}_j'$, following (Espinosa Zarlenga et al., 2022). Then, all endogenous embeddings are weighted by the strength of the dependency $a_{ij}$ and aggregated using a deepset-like neural network $f_i : \mathbb{R}^{z \times k} \to [0, 1]$ which maps endogenous embeddings to endogenous predictions:

$$\text{(endogenous embeddings)} \quad \hat{\mathbf{v}}_j' = \omega(\hat{v}_j', \hat{\mathbf{u}}_j) = \hat{v}_j' \phi_j^+(\psi(\mathbf{x})) + (1 - \hat{v}_j') \phi_j^-(\psi(\mathbf{x})) \tag{14}$$

$$\text{(endogenous variables)} \quad \hat{v}_i = f_i\left(\{a_{ij} \hat{\mathbf{v}}_j'\}_{j \in \{1,\dots,k\}}\right). \tag{15}$$

In order to learn explicit structural equations, existing logic-based aggregation methods can be used from the concept literature (Barbiero et al., 2022; 2023; Debot et al., 2024). App. F describes in more detail their adaptation in Causal CGM.

# E   UNFOLDING CGMs

Notice how learning a DAG together with Definition B.2 allows to unfold a Causal CGM's endogenous predictor applying a directed message passing on the associated structural causal model $\mathcal{M}$, ensuring that the values of endogenous variables are derived solely from the nodes that are their ancestors on the causal graph (see Figure 10). As a first step, we compute the exogenous variables for all nodes. We then predict the values of endogenous variables in root nodes in the learned DAG from their corresponding exogenous variables. Following this, we can generate the endogenous embeddings for root nodes and aggregate endogenous embeddings to compute the value of endogenous variables of each child node. We repeat this process until all leaf nodes of the graph are reached. We can obtain this by replacing in Eq. 14 the endogenous copies $\hat{v}_j'$ with the parents of the endogenous variable $v_i$:

$$\text{(unfolding)} \quad \hat{v}_i = f_i\left(\{a_{ij} \omega(\hat{v}_j, \hat{\mathbf{u}}_j)\}\right), \quad \forall i, j \in \mathcal{V} \tag{16}$$

Note that this causal unrolling guarantees two key properties (see Figure 3a and Fig. 10): (1) modifying the value of a cause (parent node) will impact the effect (child node) in our model, (2) conversely, intervening upon an effect does not alter the cause. This is because, in our model, information flows sequentially following the graph, mirroring the fundamental nature of causal effects. Consequently, this layer not only facilitates the computation of task predictions but also enables the exploration of causal relationships through do-interventions and counterfactual analysis.

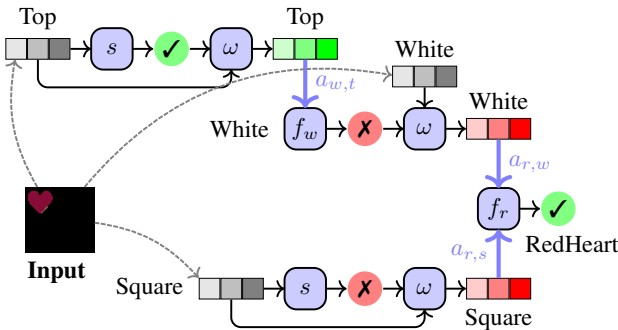

Figure 10: Unfolded Causal CGM's endogenous predictor.

In App. H.6 we explain why 1-step unfolding of the causal graph during training is equivalent to any k-step ($k > 1$) and empirically show related results.

In rare cases, some edges with very small weights may still make the adjacency matrix $A$ cyclic. At inference time, we address this by iteratively removing the edges with the smallest weights from the cycles until the graph becomes a DAG.

## F ARCHITECTURE

**Initialisation of adjacency matrix** Causal CGM provides versatile initialisation options for the adjacency matrix $A$, tailored to specific scenarios. In certain instances, weights can be derived from domain expertise or provided along with training data and labels. Without such information, weights can be directly inferred from training labels through causal structural learning algorithms (Kaddour et al., 2022) as a preliminary step. These approaches guide the model towards a predetermined decision-making pathway. Alternatively, weights can be learnt concurrently during the Causal CGM training phase, as outlined in Section 3.3. These weights may be initialised either randomly or based on the conditional entropy between labels, providing a better starting point. Additionally, a hybrid approach is feasible, where certain elements in $A$ are fixed while others remain trainable. For instance, a causal structural learning algorithm might yield a Partial Ancestral Graph (PAG) with undirected edges, allowing for the definition of directed edges and learning the direction for others to avoid cycle formation.

**Causal mechanisms** In Causal CGM, the function $f_i$ corresponds to a causal mechanism in a SCM. Such mechanisms are typically formalised via structural equations. For instance, linear models are a common choice for label predictors in Concept Bottleneck Models (Koh et al., 2020) where endogenous embeddings are aggregated using a permutation invariant aggregator function $\oplus$ (such as the element-wise maximum, or sum):

$$\hat{v}_i = \sigma \left( W_i \bigoplus_{j \in \{1,\ldots,k\}} a_{ij}\hat{\mathbf{v}}_j + \mathbf{b}_i \right) \tag{17}$$

However, other options are also available to increase the expressiveness and interpretability of the decision-making process, such as Deep Concept Reasoning (Barbiero et al., 2023) class predictors, which build logic-based formulae to obtain class label predictions using endogenous embeddings:

$$\hat{v}_i \leftarrow \bigvee_{\mathbf{x} \in X_{\text{train}}} \bigwedge_{j \in \{1,\ldots,k\}} l_j(\mathbf{x}) = \bigvee_{\mathbf{x} \in X_{\text{train}}} \bigwedge_{j \in \{1,\ldots,k\}} (\rho_{ij}(a_{ij}\hat{\mathbf{v}}_j) \iff \hat{v}'_j) \tag{18}$$

where $l_j$ denotes the literal of relevant $\hat{v}'$ representing the variable's sign or "polarity" in the logic rule (i.e., either $v_j$ or $\neg v_j$). For example, given three variables $v_1, v_2, v_3, v_4$, DCR can predict the endogenous variable $v_2$ using the rule $v_2 \leftarrow (v_1 \wedge \neg v_3) \vee (\neg v_1 \wedge v_3)$ which highlights the underlying causal mechanism linking $v_1, v_3, v_4$ to $v_2$ (notice how DCR can also learn to remove irrelevant variables such as $v_4$).

**Compositional generalisation** The training procedure of Causal CGMs is highly parallelizable and modular as only direct connections need to be trained together (e.g., $a \to b$ and $b \to c$), while the model takes care of distant connections in an indirect way. For instance, the connection $a \to b \to c$ can be obtained as a composition of two different independent training procedures for $a \to b$ and $b \to c$. As a result, it is trivial for a Causal CGM to make the causal graph grow even at test time (see Figure 11). This can be done by composing two different graphs obtained by independent training procedures, encoders, datasets, or data types. We note that this is not possible in standard CBMs, which need to re-train the task predictor from scratch whenever new concepts or tasks are added to the mix. Moreover, this modularity enables a form of out-of-distribution compositional generalisation as it creates new distant connections between variables that were never part of the same training procedure (e.g., $a$ and $c$ in the previous example).

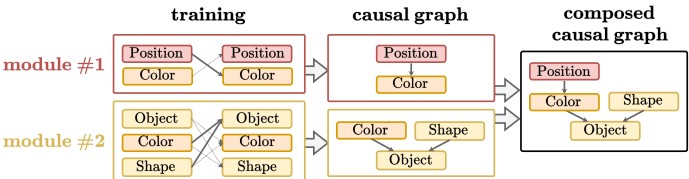

Figure 11: Compositional generalisation in Causal CGMs: two different Causal CGMs architectures are trained independently and then composed only at test time, thus creating a larger graph and allowing out-of-distribution causal inference.

## G  EXPERIMENTAL SETUP

### G.1  DATASETS

In our experiments we use three different datasets:

- **Checkmark** — The dataset consists of tabular data with three features, each ranging from $-1$ to 1 (denoted as $a$, $b$, and $c$). The target variable $d$ can be either 0 or 1. Each feature is annotated with a concept that indicates whether it is positive or negative. The dataset also incorporates causal relationships among the features. For example, feature $c$ is defined as the inverse of feature $b$. The target $d$ is set to 1 when both features $a$ and $b$ are positive. This data set is used to test our hypothesis in a straightforward and controlled setting.

- **dSprites** (Matthey et al., 2017) — The dataset comprises images featuring one of three objects (square, heart) in various positions and sizes. The defined concepts include: (1) object shape (square or heart), (2) object size (small or large), (3) vertical position (top or bottom), (4) horizontal position (left or right), (5) object colour (red or blue). Based on these, causal relationships and a binary classification task are established: if the object is a heart on the right side, it is large; if a heart is at the top of the image, it is red; the label is positive if the object is both red and large.

- **CelebA** (Liu et al., 2015) — The CelebA dataset features celebrity images annotated with various attributes, including lipstick presence, gender, facial shape, and hair type. Gender is used as the classification label. This dataset is chosen for the presence of correlations and biases, such as the association between wearing lipstick and being identified as female.

- **CIFAR-10** (Krizhevsky, 2009) — The CIFAR-10 dataset consists of 60,000 colour images in 10 different classes, such as cats, dogs, and cars, with no pre-defined concepts. To extract concepts, we follow the Label-Free Concept approach described in Oikarinen et al. (2023), where the concept set is extracted using auxiliary Vision Language Models (VLS), which are then used to establish causal relationships and perform classification tasks. This dataset is used to test our model's ability to handle complex visual features in a setting where concepts are not obtained through human annotations.

### G.2  BASELINES

We evaluated our approach against three established baselines:

- **Black Box**: This model employs a single predictor that processes the input to simultaneously predict the task label and all relevant concepts. It lacks interpretability and does not differentiate between the importance of task labels and concepts.

- **Concept Bottleneck Model (CBM) (Koh et al., 2020)**: This model first uses a concept predictor to infer concepts from the initial input, followed by a task label prediction based on these concepts. It is designed to be interpretable and treats concepts as significant informational to predict the task label.

- **Concept Embedding Model (CEM) (Espinosa Zarlenga et al., 2022)**: Comprising $n$ context encoders, one for each concept, this model predicts each concept based on its respective context before predicting the final task label. It treats concepts and task labels in the same way as CBM.

In our experiments, all baselines are trained using a joint training technique commonly used in CBMs (Koh et al., 2020). In joint training, the model is trained end-to-end with the task predictor directly taking the concept encoder's outputs as input, enabling simultaneous optimization of both concept and task predictions.

## G.3 EXPERIMENTS

In our experiments, we evaluate our approach by examining four key dimensions: (i) performance accuracy, (ii) influence of ground-truth interventions, (iii) identification of causal structures, and (iv) blocking for the influence of one variable on another.

To evaluate the first dimension, we conducted a comparative analysis of our approach (using both a learned and a predefined graph) against Black Box, CBM and CEM. This was to determine if graph-based inference would decrease model performance. For this assessment, we calculated the model's accuracy in predicting all concepts and the task. Typically, these metrics are calculated independently; however, in our study, we treated tasks and concepts equivalently, considering them collectively as labels.

In the second aspect, we evaluate our approach by comparing it against CBM and CEM in terms of response to ground-truth interventions. Enhancing the impact of interventions in the Concept-Based Model is crucial for improving the role of humans in the loop. In our experiments, we initially perturbed the inputs to reduce label prediction accuracy, following methodologies established in prior research (Espinosa Zarlenga et al., 2022). Subsequently, we implemented interventions on the most inaccurately predicted concepts in CEM and CBM. This intervention strategy is considered highly effective, as noted in (Shin et al., 2023). For the Causal CGM, interventions began with concepts that have a higher number of descendant nodes in the model's graph, aiming to maximise the intervention's effectiveness. To assess this dimension, we measured the change in accuracy for non-intervened labels before and after the interventions on $n$ concepts (Delta Label Accuracy).

In the third aspect, we visualise the DAG utilised during the inference stage by Casual CEM and derive the corresponding logic equations. We generate Sum of Product logic rules from a table that lists all possible combinations of input concept values alongside the most frequent prediction for each combination derived from the training set, similarly to what done by (Ciravegna et al., 2023). It is crucial to note that while these logic rules are general for the model decision-making process, exogenous information may alter predictions for particular instances.

In the final dimension of our analysis, we compare Causal CGM, which operates on a specified graph, against CBM and CEM in terms of their efficacy in mitigating a variable's influence on the task. Specifically, we perturb the prediction of a concept and then, following the graph structure utilised by Causal CGM, we intervene with the ground truth label on all descendant nodes (blocking). Consequently, in Causal CGM, all links between the altered concepts and the labels are deleted, a feat unachievable in CBM and CEM. To assess this characteristic, we calculated the Residual Concept Causal Effect, ratio of the Concept Causal Effect (Goyal et al., 2020) post- and pre-application of the blocking techniques. Ideally, this ratio should be zero, indicating that after blocking, the altered node's value no longer influences the outcome of the task.

### G.4 IMPLEMENTATION DETAILS

**Additional details** To maximise the efficacy of interventions in Causal CGM, the second term of the loss can be regularised to maximise the average Causal Concept Effect (CaCE) (Goyal et al., 2020) as follows:

$$\mathcal{L}_{\text{CaCE}} = \frac{1}{n} \sum_{i=0}^{n} |p(v_i|do(v_{i,r} = 1) - p(v_i|do(v_{i,r} = 0)|$$

Here, $r$ represents a randomly chosen index for each sample, which is used to select one of the concepts following Espinosa Zarlenga et al. (2022). This regularisation can be weighted using an hyperparameter, $\lambda_3$. Moreover, for all the experiments where the graph is learnt end-to-end, we initialise the learnable adjacency matrix with the conditional entropy between each pair of values, extracted from the training set.

**Hyperparameters** All baseline and proposed models were trained for varying epochs across different datasets: 500 for Checkmark, 200 for dSprites, 30 for CelebA and 25 CIFAR10. The optimal epoch for each was determined based on label accuracy on the validation set. A uniform learning rate of 0.01 was applied across all models and datasets. For the CBM and CEM models, both concept and task losses were equally weighted at 1. This weighting scheme was also applied to the loss terms for endogenous copies' prediction, endogenous variables' prediction ($\lambda_1$), and graph priors ($\lambda_2$). The weight assigned to the loss terms in our models to maximise CaCE is 0.05. Additionally, $\gamma$ was treated as a learnable parameter, initialised at 0.1, and $\beta$ was set to 1. All experiments were conducted using five different seeds (1, 2, 3, 4, 5).

**Code, licenses and hardware** For our experiments, we implement all baselines and methods in Python 3.9 and relied upon open-source libraries such as PyTorch 2.0 (Paszke et al., 2019) (BSD license), PytorchLightning v2.1.2 (Apache Licence 2.0), Sklearn 1.2 (Pedregosa et al., 2011) (BSD license). In addition, we used Matplotlib (Hunter, 2007) 3.7 (BSD license) to produce the plots shown in this paper. Two datasets we used are freely available on the web with licenses: dSprites (Apache 2.0) and CelebA, which is released for non-commercial use research purposes only. We also introduce the Checkmark dataset and we described it in this section. We will publicly release the code with all the details used to reproduce all the experiments under an MIT license. All the experiments except the CIFAR10 ones were performed on a device equipped with an M3 Max and 36GB of RAM, without the use of a GPU. The CIFAR10 experimentswere conducted on a workstation equipped with an NVIDIA RTX A6000 GPU, two AMD EPYC 7513 32-Core processors, and 512 GB of RAM.Approximately 60 hours of computational time were utilised from the start of the project, whereas reproducing the experiments detailed here requires only 8 hours.

## H ADDITIONAL RESULTS

In this section, we include all the experiments shown in Section 4 for the four datasets in more detail and an ablation study on the value of $\lambda_3$.

### H.1 GROUND-TRUTH INTERVENTIONS

Figure 12 illustrates the performance comparison among Causal CGM (using both the provided and learnt graphs), CBM, and CEM regarding the effects of interventions. Delta Label Accuracy, which quantifies the change in label accuracy before and after interventions on a growing number of concepts, is calculated solely for the concepts not directly intervened upon. In particular, Causal CGM demonstrates superior performance when interventions involve fewer concepts. This superior performance is attributed to the propagation of intervention effects through all descendant nodes in Causal CGM, unlike CBM and CEM, where the impact is confined to the final task without adjustments to other concepts. The most significant performance gain, approximately 15 percentage points, is observed in CelebA after seven interventions. This effect is particularly pronounced in scenarios with multiple concepts, such as in CIFAR10 and CelebA. In contrast, for simpler tasks with fewer concepts, like Checkmark, the benefits of our approach are less pronounced. The gap observed in CIFAR-10 between CBM and Causal CGM after all interventions is partly due to our evaluation setup: when perturbing the input prior to intervention, we occasionally generate out-of-distribution (OOD) concept embeddings, which only affected the CIFAR10 experiment. This reduces

the effectiveness of interventions for embedding-based models, while scalar-based interventions in CBMs remain unaffected. Despite this challenge, our model still achieved approximately 15% higher accuracy overall, with fewer required interventions. Furthermore, when focusing solely on the effects of interventions on the task label, the causal graph utilised by Causal CGM allows us to identify beforehand the specific subset of concepts influencing the task prediction. This pre-identification significantly decreases the required number of interventions to achieve the desired outcome. Figure 13 demonstrates that Causal CGM attains comparable improvements in task performance but after interventions on only three or four concepts, in contrast to the ten and eleven concepts required by CEM and CBM, respectively. The elevated standard error observed in Causal CGM with the learnt graph is attributed to the variability of the graph structure, which significantly influences the outcomes of interventions.

## H.2 CAUSAL STRUCTURES

Figure 14 illustrates the adjacency matrices corresponding to the DAGs used by Causal CGM for inference in the first three datasets. On the other hand, Tables 4, 5, and 6 present the logic rules derived from the adjacency matrices depicted in the aforementioned figure. Notably, in the Checkmark dataset, both configurations successfully identified the ground truth graph and the correct logic rules. In the case of dSprites, the DAG identified through causal structural learning (GRaSP (Lam et al., 2022)) accurately discovers the causal graph and associated logic rules. Although the end-to-end model accurately identifies the correct relationships between concepts and tasks, it proposes alternative methods for concept prediction. It is important to note that even though the model did not identify the correct causal graph, the model was still capable of performing causal inference with the existing graph. In the CelebA data set, where there is no ground truth for either the graph or logic rules, the findings by GRaSP and the end-to-end model appear plausible and reveal biases inherent in the dataset, such as the strong correlation between makeup use and gender or potential causal links like smiling and a slightly open mouth. This scenario underscores the benefits of employing Causal CGM, particularly in demonstrating how specific concepts are used to predict other concepts and tasks.

## H.3 ABLATION STUDY

In Tables 7, 8, and 9, we present the outcomes of varying the hyperparameter $\lambda_3$, which weights the loss term designed to enhance the CaCE effect. The results indicate that optimising this loss term contributes to improved CaCE scores, thereby augmenting the efficacy of the interventions. Nonetheless, excessively high values of $\lambda_3$ may lead to diminished model performance, as it tends to prioritise boosting the CaCE score at the expense of accurate predictions.

Table 4: Logic rules extracted for the Checkmark dataset from Causal CGM+CD with a given DAG and from Causal CGM with a learnt DAG. A term which refers to an exogenous variable is omitted for simplicity.

| METHOD | CHECKMARK |
|---|---|
| Causal CGM | $a \leftarrow \epsilon_0$ |
| | $b \leftarrow \epsilon_1$ |
| | $c \leftarrow \sim b$ |
| | $d \leftarrow a \wedge c$ |
| Causal CGM+CD | $a \leftarrow \epsilon_0$ |
| | $b \leftarrow \epsilon_1$ |
| | $c \leftarrow \sim b$ |
| | $d \leftarrow a \wedge c$ |

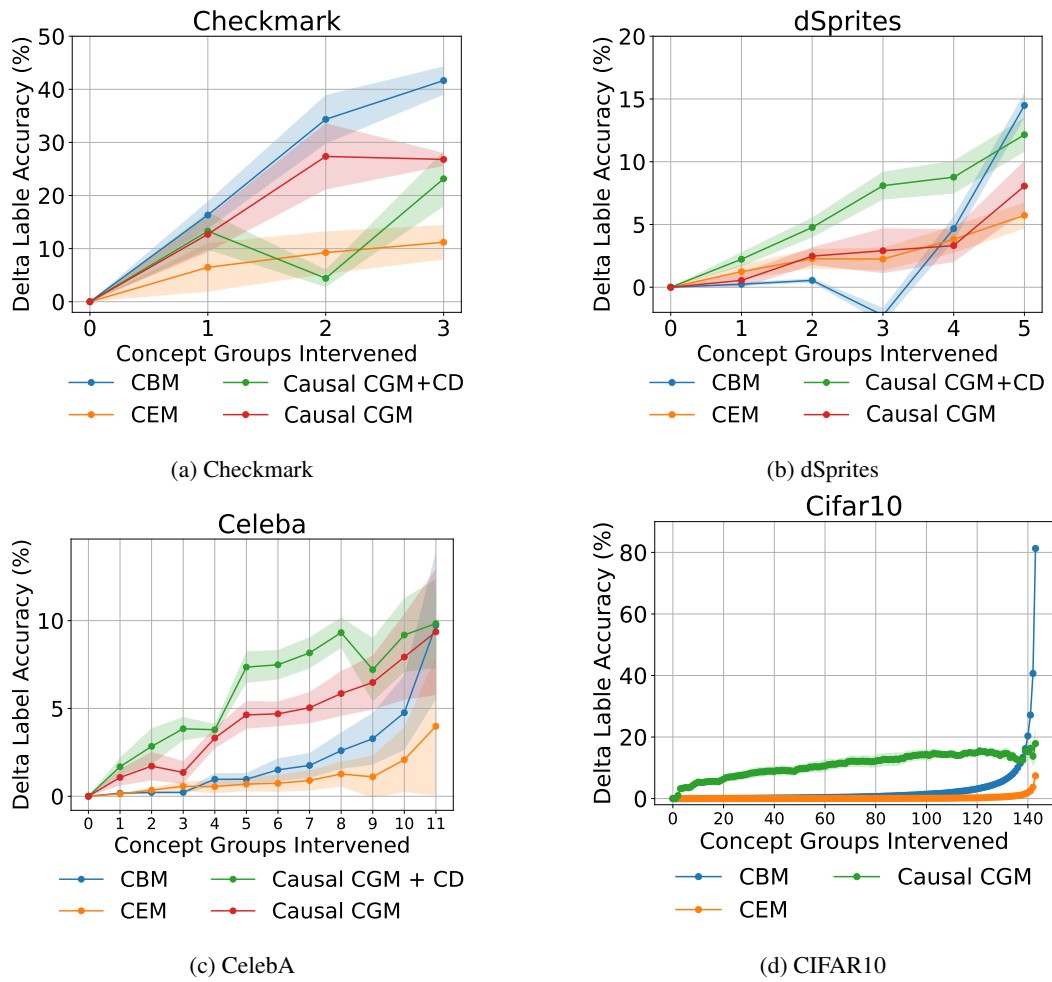

Figure 12: Impact of ground-truth interventions on concepts across four datasets. This figure illustrates the variations in accuracy for non-intervened labels, comparing performance before and after interventions on specific nodes.

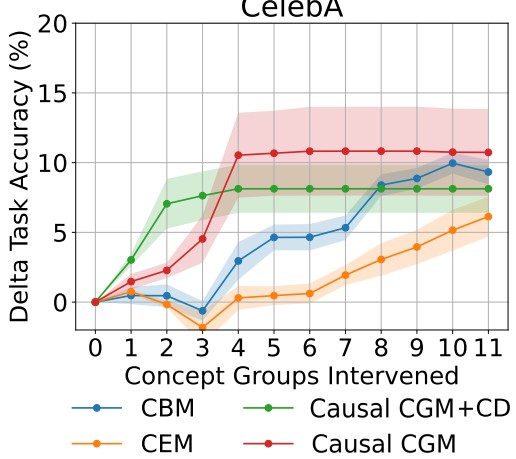

Figure 13: Impact on the task accuracy of ground-truth interventions performed on CelebA concepts. Causal CGM+CD has received in input a causal graph, discovered with a causal structural learning algorithm (GRaSP (Lam et al., 2022)), while Causal CGM learns it end-to-end.

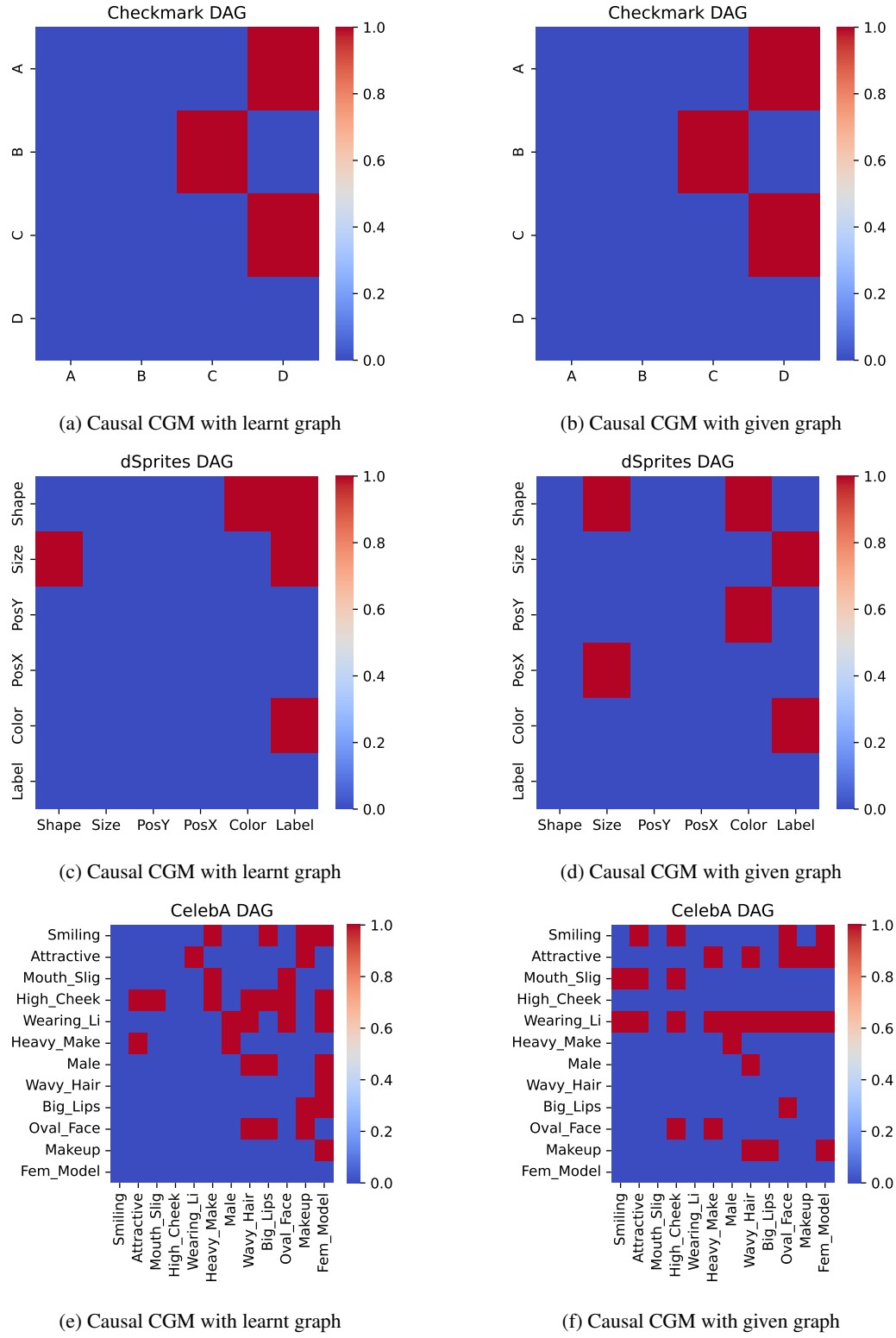

Figure 14: Adjaciency matrices representing the DAG used by Causal CGM during inference on the three datasets. On the left side, the matrices represent the DAG learnt end-to-end by the model, while on the right the DAG discovered with GRaSP (Lam et al., 2022). It provides a PAG starting from the training data.

Table 5: Logic rules extracted for the Dsprites dataset from Causal CGM+CD with a given DAG and from Causal CGM with a learnt DAG. A term which refers to an exogenous variable is omitted for simplicity.

| METHOD | DSPRITES |
|---|---|
| Causal CGM | Shape $\leftarrow$ Size |
| | Size $\leftarrow \epsilon_1$ |
| | PosY $\leftarrow \epsilon_2$ |
| | PosX $\leftarrow \epsilon_3$ |
| | Color $\leftarrow$ Shape |
| | Label $\leftarrow$ Size $\wedge$ Color |
| Causal CGM+CD | Shape $\leftarrow \epsilon_0$ |
| | Size $\leftarrow$ Shape $\wedge$ PosX |
| | PosY $\leftarrow \epsilon_2$ |
| | PosX $\leftarrow \epsilon_3$ |
| | Color $\leftarrow$ Shape $\wedge$ PosY |
| | Label $\leftarrow$ Size $\wedge$ Color |

Table 6: Logic rules extracted for the Celeba dataset from Causal CGM+CD with a given DAG and from Causal CGM with a learnt DAG. A term which refers to an exogenous variable is omitted for simplicity.

| METHOD | CELEBA |
|---|---|
| Causal CGM | Smiling (S) $\leftarrow \epsilon_0$ |
| | Attractive (A) $\leftarrow$ Heavy_Make |
| | Mouth_Slig (MS) $\leftarrow$ False |
| | High_Cheek (HC) $\leftarrow \epsilon_3$ |
| | Wearing_Li (WL) $\leftarrow$ Attractive |
| | Heavy_Make (HM) $\leftarrow$ Smiling $\wedge$ High_Cheek |
| | Male $\leftarrow \sim$ Wearing_Li $\wedge \sim$ Heavy_Make |
| | Wavy_Hair (WH) $\leftarrow$ (HC $\wedge$ WL $\wedge \sim$ Male) $\vee$ (HC $\wedge$ WL $\wedge \sim$ OF) |
| | Big_Lips (BL) $\leftarrow$ Smiling $\wedge$ High_Cheek $\wedge \sim$ Male $\wedge \sim$ Oval_Face |
| | Oval_Face (OF) $\leftarrow$ False |
| | Makeup (M) $\leftarrow$ False |
| | Fem_Model $\leftarrow$ (M $\wedge \sim$ S) $\vee$ (M $\wedge \sim$ Male) $\vee$ (M $\wedge$ HC $\wedge \sim$ WH) $\vee$ (WL $\wedge$ WH $\wedge$ BL $\wedge \sim$ S $\wedge \sim$ HC) |
| Causal CGM+CD | Smiling $\leftarrow$ Mouth_Slig |
| | Attractive $\leftarrow$ Wearing_Li |
| | Mouth_Slig $\leftarrow \epsilon_2$ |
| | High_Cheek $\leftarrow$ Smiling |
| | Wearing_Li $\leftarrow \epsilon_4$ |
| | Heavy_Make $\leftarrow$ (Attractive $\wedge$ Wearing_Li) $\vee$ (Wearing_Li $\wedge$ Oval_Face) |
| | Male $\leftarrow \sim$ Wearing_Li $\wedge \sim$ Heavy_Make |
| | Wavy_Hair $\leftarrow$ Makeup $\vee$ (Attractive $\wedge$ Wearing_Li $\wedge \sim$ Male) |
| | Big_Lips $\leftarrow$ Makeup |
| | Oval_Face $\leftarrow$ Smiling $\wedge$ Attractive $\wedge$ Wearing_Li $\wedge \sim$ Big_Lips |
| | Makeup $\leftarrow$ False |
| | Fem_Model $\leftarrow$ Makeup |

Table 7: Ablation study regarding $\lambda_3$ on the Checkmark dataset.

| | $\lambda_3$ | Label Accuracy | Average CaCE | min CaCE | max CaCE | CaCE | CaCE block |
|---|---|---|---|---|---|---|---|
| Black Box | | $90.15 \pm 0.14$ | | | | | |
| CBM | | $90.34 \pm 0.55$ | $4.09 \pm 0.80$ | $0.00 \pm 0.00$ | $9.45 \pm 1.86$ | $28.36 \pm 5.58$ | $27.79 \pm 5.64$ |
| CEM | | $89.09 \pm 1.98$ | $1.75 \pm 1.66$ | $0.00 \pm 0.00$ | $4.45 \pm 3.78$ | $11.59 \pm 12.76$ | $11.89 \pm 12.95$ |
| Causal CGM | 0.05 | $88.24 \pm 1.30$ | $14.47 \pm 2.86$ | $0.00 \pm 0.00$ | $44.37 \pm 5.58$ | $16.15 \pm 11.25$ | $0.00 \pm 0.00$ |
| Causal CGM + CE | 0.05 | $89.43 \pm 0.93$ | $13.15 \pm 2.31$ | $0.00 \pm 0.00$ | $39.53 \pm 3.62$ | $20.99 \pm 8.19$ | $0.00 \pm 0.00$ |
| Causal CGM | 0.2 | $85.88 \pm 3.31$ | $11.64 \pm 3.21$ | $0.00 \pm 0.00$ | $37.55 \pm 8.77$ | $16.32 \pm 15.36$ | $0.00 \pm 0.00$ |
| Causal CGM + CE | 0.2 | $85.09 \pm 2.78$ | $16.39 \pm 5.00$ | $0.00 \pm 0.00$ | $39.69 \pm 10.97$ | $31.15 \pm 19.48$ | $0.00 \pm 0.00$ |
| Causal CGM | 0 | $87.86 \pm 1.66$ | $6.98 \pm 2.95$ | $0.00 \pm 0.00$ | $18.16 \pm 8.29$ | $7.39 \pm 4.10$ | $0.00 \pm 0.00$ |
| Causal CGM + CE | 0 | $87.04 \pm 2.72$ | $12.16 \pm 2.92$ | $0.00 \pm 0.00$ | $33.15 \pm 9.97$ | $14.52 \pm 14.29$ | $0.00 \pm 0.00$ |

Table 8: Ablation study regarding $\lambda_3$ on the dSprites dataset.

| | $\lambda_3$ | Label Accuracy | Average CaCE | min CaCE | max CaCE | CaCE | CaCE block |
|---|---|---|---|---|---|---|---|
| BlackBox | | $99.53 \pm 0.05$ | | | | | |
| CBM | | $99.55 \pm 0.07$ | $2.77 \pm 0.30$ | $0.00 \pm 0.00$ | $8.51 \pm 1.03$ | $0.64 \pm 0.70$ | $0.64 \pm 0.70$ |
| CEM | | $99.48 \pm 0.07$ | $0.46 \pm 0.29$ | $0.00 \pm 0.00$ | $2.45 \pm 1.67$ | $2.43 \pm 4.62$ | $2.43 \pm 4.62$ |
| CausalCEM | 0.05 | $99.44 \pm 0.11$ | $17.53 \pm 3.26$ | $0.00 \pm 0.00$ | $44.47 \pm 10.16$ | $34.13 \pm 19.46$ | $0.00 \pm 0.00$ |
| Causal CGM + CE | 0.05 | $99.40 \pm 0.15$ | $12.85 \pm 0.59$ | $0.00 \pm 0.00$ | $27.72 \pm 3.66$ | $28.95 \pm 13.50$ | $0.00 \pm 0.00$ |
| Causal CGM | 0.2 | $98.80 \pm 1.24$ | $14.13 \pm 3.39$ | $0.00 \pm 0.00$ | $37.59 \pm 14.20$ | $17.52 \pm 13.41$ | $0.00 \pm 0.00$ |
| Causal CGM + CE | 0.2 | $99.30 \pm 0.13$ | $12.90 \pm 0.51$ | $0.00 \pm 0.00$ | $34.01 \pm 6.99$ | $16.84 \pm 6.79$ | $0.00 \pm 0.00$ |
| Causal CGM | 0 | $99.58 \pm 0.12$ | $6.89 \pm 1.55$ | $0.00 \pm 0.00$ | $18.51 \pm 5.14$ | $12.11 \pm 13.01$ | $0.00 \pm 0.00$ |
| Causal CGM + CE | 0 | $99.51 \pm 0.05$ | $5.67 \pm 1.21$ | $0.00 \pm 0.00$ | $12.41 \pm 1.82$ | $15.15 \pm 4.10$ | $0.00 \pm 0.00$ |

## H.4 CONCEPT INCOMPLETENESS AND GENERALISATION WITH A DIFFERENT NUMBER OF CONCEPTS

Concept incompleteness does not have a strong impact on Causal CGM. As shown in the experiments with CLIP on CIFAR, we can automatically extract new concepts if the provided ones are insufficient. However, to quantify the impact of incompleteness on classification performances, we report an ablation study on a CBM benchmark (Marcinkevičs et al., 2024) specifically designed to assess concept incompleteness. This benchmark is a synthetic tabular dataset where we can control to what degree concepts explain the variance in the downstream task (see (Marcinkevičs et al., 2024) for details). In Figure 15 we observe that, as the concept incompleteness $i$ increases from $i = 0$, where all relevant concepts are available, to $i = 0.9$, where only $10\%$ of relevant concepts are available, the performance of CBMs drop, while that of black boxes, CEMs, and Causal CGMs do not thanks to the use of concept embeddings.

## H.5 SCALABILITY AND COMPUTATIONAL COMPLEXITY

Causal CGMs pays a small cost (at most quadratic in the number of class labels) to address causal transparency. In terms of the number of parameters, the endogenous predictor scales linearly with the number of concepts, as in standard CBMs. Regarding time complexity, the endogenous predictor scales quadratically (both in train and inference) with the number of concepts. This is because, during training, we model all concept dependencies, and, during testing, inference is performed on a DAG (a triangular adjacency matrix in the worst case). This is a small cost in exchange for interpretability (and not unique to our work). To better communicate our method's scalability, we report the number of parameters and training and inference times as we increase the number of class labels on the same CBM benchmark used to assess incompleteness (Marcinkevičs et al., 2024). In Figure 16 we observe

Table 9: Ablation study regarding $\lambda_3$ on the CelebA dataset.

| | $\lambda_3$ | Label Accuracy | Average CaCE | min CaCE | max CaCE | CaCE | CaCE block |
|---|---|---|---|---|---|---|---|
| Black Box | | $90.15 \pm 1.30$ | | | | | |
| CBM | | $79.00 \pm 0.18$ | $0.54 \pm 0.03$ | $0.00 \pm 0.00$ | $1.67 \pm 0.15$ | $5.58 \pm 2.36$ | $5.46 \pm 2.13$ |
| CEM | | $79.17 \pm 0.26$ | $0.27 \pm 0.12$ | $0.00 \pm 0.00$ | $1.07 \pm 0.56$ | $1.00 \pm 0.45$ | $1.06 \pm 0.50$ |
| Causal CGM | 0.05 | $78.23 \pm 0.45$ | $2.17 \pm 1.44$ | $0.00 \pm 0.00$ | $8.18 \pm 4.38$ | $0.04 \pm 0.07$ | $0.00 \pm 0.00$ |
| Causal CGM + CE | 0.05 | $78.42 \pm 0.42$ | $5.48 \pm 0.34$ | $0.00 \pm 0.00$ | $24.17 \pm 1.32$ | $1.24 \pm 0.62$ | $0.00 \pm 0.00$ |
| Causal CGM | 0.2 | $77.49 \pm 0.37$ | $1.70 \pm 0.98$ | $0.00 \pm 0.00$ | $8.95 \pm 4.86$ | $0.00 \pm 0.00$ | $0.00 \pm 0.00$ |
| Causal CGM + CE | 0.2 | $78.08 \pm 0.39$ | $6.15 \pm 0.31$ | $0.00 \pm 0.00$ | $29.57 \pm 1.03$ | $0.82 \pm 0.46$ | $0.00 \pm 0.00$ |
| Causal CGM | 0 | $77.42 \pm 1.09$ | $2.85 \pm 0.44$ | $0.00 \pm 0.00$ | $13.06 \pm 2.65$ | $0.00 \pm 0.00$ | $0.00 \pm 0.00$ |
| Causal CGM + CE | 0 | $78.31 \pm 0.36$ | $4.64 \pm 0.13$ | $0.00 \pm 0.00$ | $18.31 \pm 2.61$ | $1.25 \pm 0.68$ | $0.00 \pm 0.00$ |

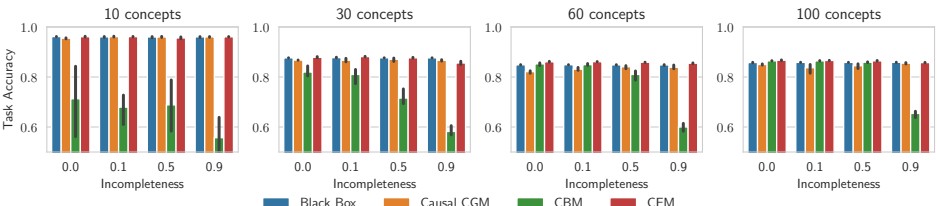

Figure 15: Label accuracy for different numbers of concepts with increasing concept incompleteness.

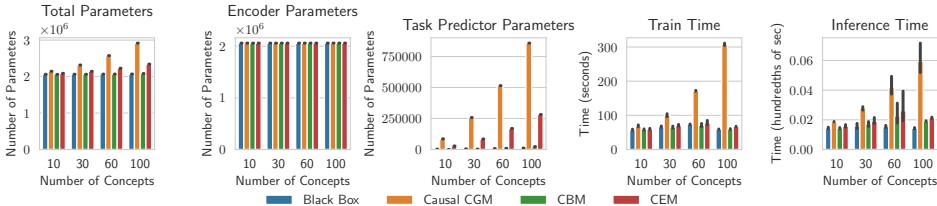

Figure 16: Number of parameters, train and inference time.

that the impact on the number of parameters is negligible as it is mostly due to the encoder. Here the encoder is a relatively small network, but it could be a billion-parameter in a Vision-lLanguage Model (VLM). Training and inference time are slightly higher than baselines, but if training includes a large encoder, say a VLM, the relative impact of the endogenous predictor would be significantly reduced (we did not have time to train a VLM from scratch).

### H.6   $k$-UNFOLDING ABLATION

In Causal CBM, 1-step unfolding of the causal graph during training is equivalent to any k-step ($k > 1$) unfolding as any step corresponds (theoretically and empirically) to repeating the same query on the same conditional probability table (CPT). Causal CGM learns the conditional probabilities of the variables given its parents (i.e. the CPT of the Bayesian network). Under full observability, conditional queries of the form $p(v|$ all but $v)$ are just inspections of the CPT and do not require any reasoning on the cyclical model (which alleviates us from setting a complex $k$-step semantics to unroll the cyclical model). The key observation is that the cyclical model served the purpose of templating multiple (any) acyclical graphs jointly without making assumptions on concept relations. During learning, the probabilities of the CPT (parameterised by NNs) are also subject to an acyclicity objective. After learning, the resulting graph is acyclic thus inference is performed via standard message passing along the directed acyclic graph. To show this experimentally, we report the results on our CBM benchmark using a different number of unfolding steps and considering different levels of concept completeness in Table 10. As expected, the results show that increasing the number of steps does not impact Causal CGM's classification performances.

| | Incompleteness ratio | | | |
| --- | --- | --- | --- | --- |
| | 0.0 | 0.1 | 0.5 | 0.9 |
| Unfolding K=1 | $85.00 \pm 0.14$ | $83.70 \pm 1.10$ | $84.49 \pm 0.57$ | $85.51 \pm 0.16$ |
| Unfolding K=2 | $83.78 \pm 0.81$ | $83.72 \pm 0.85$ | $85.26 \pm 0.20$ | $84.96 \pm 0.14$ |
| Unfolding K=3 | $85.00 \pm 0.34$ | $83.10 \pm 0.98$ | $84.30 \pm 0.47$ | $85.32 \pm 0.05$ |

Table 10: Causal CGM's accuracy with $k = \{1, 2, 3\}$-unfolding and higher concept incompleteness.

## H.7 DEBIASING VIA DO-INTERVENTIONS

We can measure the effectiveness of blocking in debiasing Causal CGMs using the following procedure:

- Selection of Variables $i$ and $j$: We randomly select a pair of concepts, $i$ (parent) and $j$ (child), connected in the graph discovered by the CD algorithm or CausalCGM.

- Do-Intervention: A do-intervention is applied to the child node ($j$) to cut off its dependence on the parent node ($i$). In our model, this intervention ensures that all subsequent nodes in the graph become independent of the parent node, as information flow is blocked by design.

- Measuring the Impact: After the intervention, we modify the value of the parent node ($i$) and measure its impact on the final label (leaf node) with $i$ as an ancestor. If no effect is observed, it indicates successful debiasing, as expected by our model's design. Cutting the edge breaks the information flow through the graph—from root to leaf—preventing the parent node from influencing subsequent nodes.

- Comparison with Baselines: Baselines have no graph constraints, therefore the do-intervention modifies the value of a concept but does not alter the decision-making process. As a result, the parent node's influence cannot be entirely removed. This contrasts our model's ability to block the information flow fully.

The results of this experiment are reported in Table 3 using the Residual Concept Causal Effect ($\downarrow$) between causally-related variables having blocked all paths between the two variables with do-interventions on the causal graph. The optimal value is zero corresponding to perfect causal independence. To explain why the optimal value is zero, consider the following example. Consider a graph $A \rightarrow B \rightarrow C$. For this graph, the variable $A$ is a cause of the variable $C$. As a result, we expect the average causal effect of $A$ on $C$ to be higher than $0$. Our experiment aims to show that intervening on $B$ makes $A$ and $C$ causally independent (cf. with L323-331). To show this, we apply a do-intervention in $B$, removing the edge $A \rightarrow B$ and generating two disconnected graphs: $A$ and $B \rightarrow C$. Now, we expect that the residual average causal effect of $A$ on $C$ is $0$ because there is no longer a connection between $A$ and $C$. As a result, $0$ is the optimal desirable outcome.

# I ANSWERING COUNTERFACTUAL QUERIES WITH CAUSAL CONCEPT GRAPH MODELS

In our experiments, we use Causal CGMs to compute Pearl's Probability of Sufficiency and Necessity (PNS), which is a prototypical example of a counterfactual query (Tian & Pearl, 2000). In general, the computation of counterfactual queries requires three key ingredients (Pearl, 2009): (i) a causal graph, (ii) structural rules predicting an endogenous variable given its parents' values, and (iii) information about the exogenous variables distribution, which for us is encoded in the embeddings. In causal CGM, all these three ingredients are available, allowing the computation of both identifiable and unidentifiable causal queries. For the latter, as is the case of PNS, the result is given in terms of an interval exactly as it happens in causal inference with structural causal models (with probabilistic quantification over exogenous variables). Figure 17 illustrates the upper boundaries of the PNS tables obtained using Causal CGM across the four datasets. Additionally, Table 11 provides an example of counterfactuals generated on the CelebA dataset, where we first intervene on the lipstick concept, followed by an intervention on the makeup concept.

Table 11: Examples of counterfactuals generated on CelebA obtained via do-interventions (intervened variables are marked in red, changed variables are underlined).

| | Endogenous variables' activations |
|---|---|
| Initial Predicted State | Smiling, Not Attractive, Mouth Slig, High Cheek, Not Wearing Li, Not Heavy Make, Not Wavy Hair, Not Big Lips, Not Oval Face, Not Makeup, Not Fem Model |
| What if I would wear lipstick? | Smiling, Attractive, Mouth Slig, High Cheek, Wearing Li, Heavy Make, Not Wavy Hair, Not Big Lips, Not Oval Face, Not Makeup, Not Fem Model |
| What if I would wear lipstick and also makeup? | Smiling, Attractive, Mouth Slig, High Cheek, Wearing Li, Heavy Make, Not Wavy Hair, Big Lips, Not Oval Face, Makeup, Fem Model |

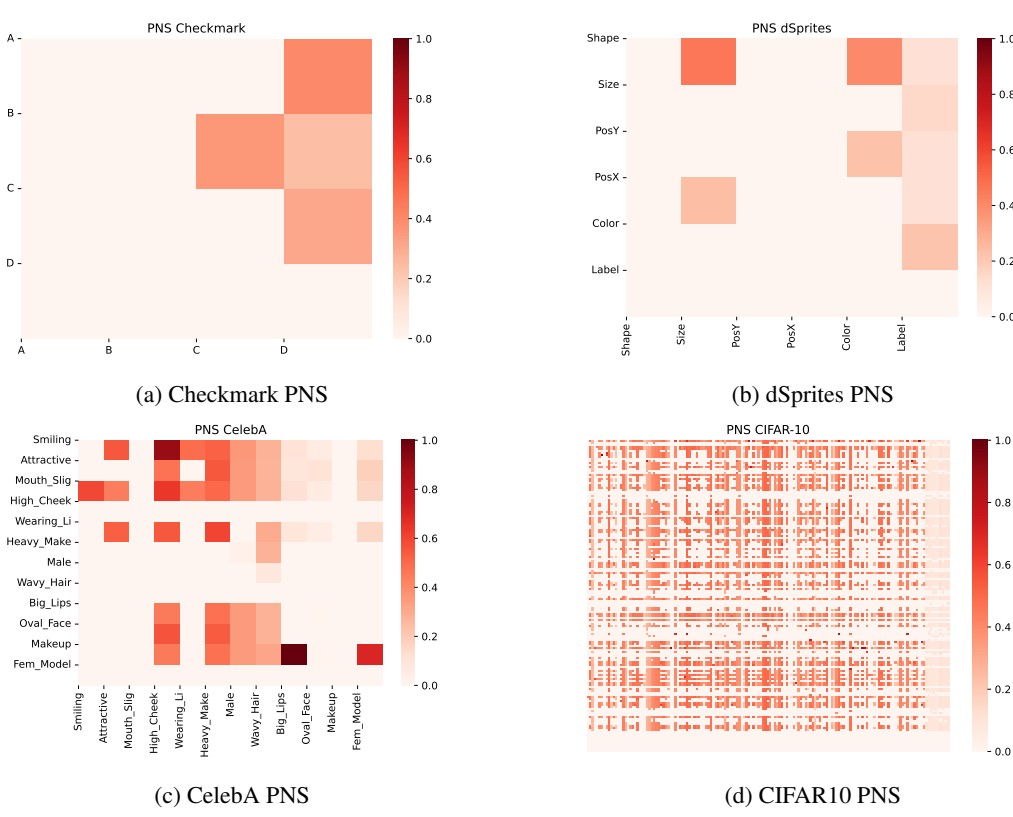

(a) Checkmark PNS

(b) dSprites PNS

(c) CelebA PNS

(d) CIFAR10 PNS

Figure 17: The upper boundary of the PNS computed on the Causal CGM models on the four different datasets. The PNS is computed between each couple of nodes that are directly or indirectly connected in the graph.

## J  CAUSAL CONCEPT GRAPH MODELS ADDRESS "CAUSAL OPACITY", NOT "CAUSAL DISCOVERY"

Our work is based on a fundamental yet subtle distinction between causal opacity/transparency and causal discovery that is poorly understood in the ML literature. Although related, the two problems are different in nature: Causal opacity/transparency refers to the problem of understanding and manipulating the causal structure of a model's decision-making process, and it thus is a specific instance of what is usually referred to as the opacity or black-box problem. Causal discovery, on the other hand, is not concerned with how the model is but rather with how the world (Boge, 2022), i.e., it concerns determining the causal structure of the generating mechanism beyond data. This distinction applies also to the notion of causal understanding. Indeed, we can distinguish between two forms of causal understanding depending on whether the object of one's understanding is the model's decision-making process or the world/data-generating mechanism. Notice that causal opacity (and the related form of causal understanding) neither implies nor requires causal discovery. The causal structure of the model's decision-making process does not necessarily mirror the causal structure of the world. Of course, one could argue that a good model should always resemble the world's causal structure, but this is not always true or even wanted (e.g., if the world is biased, we may want to construct an unbiased model that deviates from the data-generating process of its training data).

## K  RELATIONS WITH CAUSAL REPRESENTATION LEARNING

If we assume a strict definition of causal representation learning (CLR) as any method that aims at learning high-level representations of the causal-generating mechanisms of data (Schölkopf et al., 2021), then Causal CGMs are not an example of CRL. Causal CGMs aims not to discover causal mechanisms beyond data but to build models whose decision-making processes follow a causal structure that users can visualize, interpret, and manipulate. This requires the model to use high-level variables (concepts) that are meaningful for a user (as opposed to, say, an image's pixels) and to clearly display the causal relationships between these variables (e.g., via a casual graph).

Regardless, there are similarities between Causal CGMs and CRL. Both rely on Pearl's formalism of SCMs and use high-level interpretable variables and causal graphs to model causal dependencies. In the case of Causal CGMs, however, the learned causal structure may not necessarily mirror the causal structure of the data-generating mechanism, as it is required for CRL models.

## L  COMPARING DIFFERENT DECISION MAKING PROCESSES WITH CAUSAL CONCEPT GRAPH MODELS

Causal CGMs are not identifiable in the sense that we can obtain different causal graphs by training the model over the same dataset. This is similar to obtaining different parameter values and embeddings when training standard deep-learning black boxes on the same data. Similarly, the training of causal CGMs can result in different causal decision-making structures for the same task. The difference is that in the case of the black box, we cannot compare the different models because they are (causally) opaque. In the case of Causal CGMs, we can inspect the multiple models obtained and verify them against desirable properties (e.g., fairness), hence adopting the best one. This distinction impacts key properties, including identifiability. To further clarify this point, consider the following example. In the world, there is a single ground truth causal structure (e.g., "smoking causes lung cancer"). Guaranteeing identifiability is crucial here because it ensures that we can reliably recover the ground truth causal graph. In contrast, causal opacity deals with DL models, which are systems that can implement a variety of decision-making mechanisms (e.g., a valid decision mechanism for a DL model is: "the model predicts that an individual has lung cancer and this causes the model to predict that the individual has a smoking habit"). These mechanisms may differ even when trained on the same data, as DL models can learn different representations to predict the same outputs. As a result, there is no single "ground truth" causal structure to identify—multiple valid causal graphs exist, each reflecting a different way the model processes information. This inherent variability makes identifiability irrelevant in the context of causal opacity.

