# OpenReview forum: "Causal Concept Graph Models: Beyond Causal Opacity in Deep Learning"
_ICLR.cc/2025/Conference — ICLR 2025 Poster_

### Official Review · Reviewer_JdcH · 2024-10-17

**Soundness:** 4
**Presentation:** 3
**Contribution:** 1
**Rating:** 6
**Confidence:** 2

**Summary:**

The authors introduce a generalization of *Concept Bottleneck Models* (CBM), called *Causal Concept Graph Models* (causal CGM). In contrast to existing CBMs, causal CGM allows for modeling causal relationships between latent variables. In contrast to causal representation learning, causal CGM aims at addressing causal opacity (interpretability) of deep learning models, rather than identifying specific variables and/or mechanisms. The claimed properties of Causal CGM are experimentally assessed on various data sets and compared with a black-box deep learning baseline, a non-causal CBM baseline and Concept Embedding Models.

**Strengths:**

1. The authors present an interesting perspective on causal representation learning, where the learned abstract variables and mechanisms are used to obtain insights into a trained model (interpretability) rather than aiming at generalizing to unseen domains. The latter is the main motivation of the related work that I am familiar with (e.g., [1, 2, 3]).

2. The manuscript is particularly well-written and the plots are great, too. The authors did an excellent job at presenting their work well.

3. There are plenty of experiments that (somewhat surprisingly) show that causal CGMs perform similarly to black box models that do not enforce a causal bottleneck.

[1] von Kügelgen, Julius, et al. "Nonparametric identifiability of causal representations from unknown interventions." Advances in Neural Information Processing Systems 36 (2024).

[2] Lippe, Phillip, et al. "Biscuit: Causal representation learning from binary interactions." Uncertainty in Artificial Intelligence. PMLR, 2023.

[3] Mengyue Yang, Furui Liu, Zhitang Chen, Xinwei Shen, Jianye Hao, and Jun Wang. Causalvae: Structured causal disentanglement in variational autoencoder. arXiv preprint arXiv:2004.08697, 2020.

**Weaknesses:**

1. In terms of the architecture and training of the proposed method, I am concerned about novelty. The method may seem novel within the CBM literature, where only independent latent variables have been considered so far. However, the proposed method seems to be (almost) identical to [1], which is branded as *causal variational autoencoder* rather than *concept bottleneck model*. Apart from the name and problem motivation, I cannot identify any differences. It would be great if the authors could highlight the differences, if they exist (I did not see them).

2. While the authors start with an interesting perspective on learning latent causal variables (see strength 1), it is unfortunate that this idea is not taken very far. Specifically, I would be interested in questions such as how such a causal graph could be efficiently inferred from any black-box model post training (just an example of the kind of questions that came up to me). Instead, causal CGM seems to be an existing causal representation learning method (see weakness 1), but without identifiability guarantee. I would advise the authors to develop the paper more closely along this original idea (strength 1), as I believe there is a lot of unused potential.

3. The *human-in-the-loop corrections* aspect seems contradictory to the motivation of the work: Causal CGM is introduced from the point of interpretability rather than identifiability of variables and mechanisms. Human-in-the-loop corrections indicate that specific variables (known the human) ought to be identified, which lies at odds with the causal opacity motivation. Therefore, I would suggest moving this aspect entirely to the appendix.

[1] Mengyue Yang, Furui Liu, Zhitang Chen, Xinwei Shen, Jianye Hao, and Jun Wang. Causalvae: Structured causal disentanglement in variational autoencoder. arXiv preprint arXiv:2004.08697, 2020.

**Questions:**

line 88: (Minor comment) The authors write the causal CGM is related to XAI. Maybe it is more about interpretability than it is about explainability? To me, XAI methods aim at explaining black-box models. However, causal CGM is rather about constructing the model such that it is interpretable by construction.

line 91: As already explained in weakness 1, I am not convinced that causal CGM is a new architecture.

line 105: I would challenge that structural causal models are *the* standard framework for causal modeling. Rubin causal models are also quite common. I would suggest rephrasing this. For example: *"Structural causal models are a widely used framework for causal modeling, alongside other approaches like Rubin causal models."*

line 111: *"which determine the values of each endogenous variable $v_i \in V$ by computing the conditional probability ..."* This sentence sounds contradictory to me. According to the SCM definition that I am familiar with, all endogenous variables can be expressed as a deterministic function of all exogenous variables (reduced form expression). Thus, this conditional probability should be just a Dirac measure? I would appreciate if the authors could clarify this.

line 115: I believe this should read *hard interventions*. There also exist other types of interventions where variables are not fixed to a value.

line 121: I am not too familiar with the literature on *concept-based models* (CBM). However, I do not understand how the term *concept* differs from *high-level feature* and how *concept-based modeling* is distinct from *representation learning* (see weakness 2).

line 199: (Minor comment) Doesn't theorem 3.2 follow from d-separation? I could imagine that the proof could be greatly simplified with such a graphical argument.

line 245: How does the training differ from the method proposed by [1] (see weakness 1)? It would be great if the authors could highlight this in detail.

line 257: *maximise the log-likelihood of v, v′* How can the likelihood of unobserved variables be maximized? I suppose you maximize a variational lower bound of $\text{log} \\; p(x)$?

line 481: *Causal CGMs focus on high-level human interpretable concepts* As mentioned in weakness 3, this seems to be at odds with the original motivation of the work.

---

> ### Author Response · Authors · 2024-11-22
> **Response to Reviewer JdcH (1/2)**
>
> ### Relation with causal variational autoencoder
> **The training and loss are similar, but the model, the architecture, and the inference at test time are not**:
>
> **1. In Causal VAE, concepts are scalars, while in Causal CGMs, concepts are high-dimensional supervised embeddings that are much more expressive (as detailed in App D).**
>
> **2. CGMs’ causal transparency is a property that derives from the test-time unrolling. Causal VAE does not perform test-time unrolling**. Also, Causal CGM unrolling will not be effective without the expressivity of concept embeddings (see below ablation result).
>
> **3. Causal VAE are generative models, CGMs are discriminative**
>
> The main similarity is that Causal CGMs use the same DAG loss of Causal VAE (we cited it at L245). So the training is similar, but the problem, the model, the architecture, and the inference at test time are not. Regarding point 2. see ablation study using scalar instead of embeddings in the common answer.
>
> ---
>
> ### Human-in-the-loop corrections aspect seems contradictory to the motivation of the work
> **The effectiveness of human-in-the-loop corrections aims to assess whether the learned causal mechanisms are aligned with humans’ expectations**. Also, we assume concepts are not latent variables to discover, which may or may not align with human concepts. We assume that concepts are supervised by humans, thus they are aligned with human semantics as in CBMs. In extreme cases (CIFAR10), we use a VLM to label concepts to reduce the cost of producing concept labels, but the label's meaning still needs to be aligned with human semantics. What may not be aligned are the causal relations among the concepts used by the model and by a human for reasoning. However, our experiments show two things: 1) Causal CGMs’ interventions are much more effective than in existing CBMs, 2) when combined with causal discovery techniques, Causal CGMs’ inference aligns possibly better with the data generating process and human inference structure making interventions even more effective showing the benefit of combining Causal CGMs with inductive biases improving semantic alignment and generalization ability (such as, but not limited to causal discovery techniques).
>
> ---
>
> ### Causal CGM is more related to interpretability rather than XAI
> **We agree.** This is why we described our method as “causally transparent by design” (L92).
>
> ---
>
> ### Rubin causal models
> **Thanks for pointing this out!** We have rephrased this in our new manuscript as: “\emph{Structural Causal Models} (SCM) proposed by \citet{Pearl2009} represent one of the possible frameworks for causal modelling”.
>
> ---
>
> ### Reduced form expression
> In theory, the conditional distribution in L111 is degenerate because endogenous variables can be expressed as deterministic functions of the exogenous variables. However, from a Bayesian perspective, if we consider that the parameters of the structural equations might also be uncertain, then the conditional distribution may no longer be degenerate. In this work we did not consider such a Bayesian perspective, but this could be an option for future works.
>
> ---
>
> ### Hard interventions
> **Agree. Rephrased as “hard interventions”**
>
> ---
>
> ### Concepts (concept-based learning) vs High-level features (representation learning)
> **In representation learning, there is no semantic requirement that high-level features are aligned to human semantics, while in concept learning, this is a hard requirement.** For instance, the embedding of a ResNet may represent high-level units of information, but this does not mean that this information has a semantic correspondence with a concept humans would use to reason about the problem at hand (if anything, this is very unlikely and there is evidence of superposition). This semantic alignment is necessary for human interventions to be effective, and it is a core requirement of all CBM-like architectures as discussed in [1].
>
> ---
>
> ### Doesn't theorem 3.2 follow from $d$-separation?
> The proof uses some conditional independence properties of the distribution (similarly to the $d$-separation criterion), but the proof is specific to our PGM where the two sets of variables $V$ and $V^\prime$ are identical copies of the same endogenous variables.
>
> ---
>
> ### How can the likelihood of unobserved variables ($v$, $v{\prime}$) be maximized?
> **$V$ and $V^\prime$ are identical copies of *observed* variables (L256).** Hence, the maximisation of the log-likelihood leads to the minimization of the standard cross-entropy loss between the model’s prediction and the ground-truth labels.
>
> ---
>
> [1] Kim, Been, et al. "Interpretability beyond feature attribution: Quantitative testing with concept activation vectors (tcav)." International conference on machine learning. PMLR, 2018.

---

> > ### Author Response · Authors · 2024-11-22
> > **Response to Reviewer JdcH (2/2)**
> >
> > ### How a causal graph could be efficiently inferred from any black-box model post-training
> > In related works (L524-30), we discuss how there are methods to build surrogate causal models to emulate a target black box model’s predictive behaviour (as in [2]). However, as Rudin [3] notes, convergence in predictive behaviour does not ensure structural similarity in decision-making processes, making surrogate models questionable as explanatory proxies. Causally interpretable by-design architectures, such as Causal CGMs, do not suffer from this issue.
> >
> > ---
> >
> > [2] Moreira, Ricardo, et al. "DiConStruct: Causal Concept-based Explanations through Black-Box Distillation." arXiv preprint arXiv:2401.08534 (2024).
> >
> > [3] Rudin, Cynthia. "Stop explaining black box machine learning models for high stakes decisions and use interpretable models instead." Nature machine intelligence 1.5 (2019): 206-215.

---

> > > ### Comment · Reviewer_JdcH · 2024-11-26
> > >
> > > I thank the authors for the comprehensive response. I would like to make some remarks.
> > >
> > > > 1. In Causal VAE, concepts are scalars, while in Causal CGMs, concepts are high-dimensional supervised embeddings that are much more expressive (as detailed in App D).
> > >
> > > I agree that this is a difference to Causal VAE. However, this modification does still not present a novelty. Such multi-dimensional variables have been considered in causal representation learning before (see e.g., [1]).
> > >
> > > > Causal VAE does not perform test-time unrolling.
> > >
> > > I am not sure if I understand "test-time unrolling" correctly. My understanding is that this means that you plug in all exogenous variables into the SCM to compute all endogenous variables. Why can this not be done using Causal VAE?
> > >
> > > > Causal VAE are generative models, CGMs are discriminative
> > >
> > > Ok, thanks. My interpretation was that the method first learns the latent variables in an unsupervised fashion, and then these learned variables are used as predictors for some target variable. Looking at the manuscript again, I must admit that I still do not understand how this model is trained. It seems the manuscript addresses an audience that is familiar with CBMs and I am not. The authors write that these high-dimensional concepts are given. Frankly, I cannot imagine a single practically relevant example where this is the case. In most practical settings, one typically only has access to the $x$ variable (i.e., a high-dimensional image) and nothing else (besides maybe a few scalar annotations).
> > >
> > > >  In representation learning, there is no semantic requirement that high-level features are aligned to human semantics
> > >
> > > I am not sure if I can agree with this statement. I do not know how "human semantics" are defined, but typically a generative model is assumed in causal representation learning where the "underlying factors of variation" are interpretable to humans. By the way, interpretability is one of the core arguments of the causal representation learning community (see e.g., [2]).
> > >
> > > I greatly appreciate the authors' efforts in addressing my concerns. All in all, however, I am still not convinced of the manuscript. Given that I am presumably not the target audience of the work and I do not understand some parts of the manuscript, I will decrease my confidence. However, I will remain at my score.
> > >
> > > [1] Lippe, Phillip, et al. "Citris: Causal identifiability from temporal intervened sequences." International Conference on Machine Learning. 2022.
> > >
> > > [2] Rajendran, Goutham, et al. "Learning interpretable concepts: Unifying causal representation learning and foundation models." arXiv preprint arXiv:2402.09236 (2024).

---

> > > > ### Author Response · Authors · 2024-12-04
> > > >
> > > > We thank the reviewer for their detailed and constructive feedback. We address the last points raised as follows:
> > > >
> > > > *1. Intervening on high-dimensional embeddings:*
> > > >
> > > > In our approach, we can intervene directly on the high-dimensional embeddings (taking inspiration from [1]) by modifying the scalar values they represent. This is a significant distinction, as high-dimensional embeddings typically do not represent interpretable variables.
> > > >
> > > > By contrast, in our method, the scalar values associated with these embeddings enable interpretability and controlled interventions.
> > > >
> > > > *2. Concept labels and supervision:*
> > > >
> > > > Concept labels, represented as scalar values, are given in our framework. While these scalar labels are provided in some settings, they can also be extracted automatically, as demonstrated in the CIFAR10 experiment.
> > > >
> > > > Importantly, we do not supervise the high-dimensional embeddings themselves but rather the scalar values they represent, allowing for a more interpretable learning process.
> > > >
> > > > *3.Training and alignment in the loss function:*
> > > >
> > > > As shown in the loss function, we optimize using cross-entropy (CE) for both the variable and its copy. The scalar or binary labels serve as supervision, ensuring that the variable and its copy are aligned to the same label. This guarantees consistency between the embeddings and their scalar representations.
> > > >
> > > > *4. “Features aligned with human semantics”:*
> > > >
> > > > Here by "features aligned with human semantics" we mean features that are understandable by a human user and that a human user can use in its reasoning (e.g., color, shape, ...). Notice that we refer here to representation learning in general, and not to causal representation learning: specific interpretability requirements hold only for this specific subfield. The underlying factors of variations in causal representation learning are indeed concepts. However, concept-based modeling must be kept distinct also from causal representation learning, as in concept-based modeling the goal is to pass information through a filter of variables interpretable by humans to make predictions more transparent, the goal of causal representation learning is to identify causal-generating mechanisms.
> > > >
> > > > We hope this clarifies the distinctions and addresses the concerns raised by the reviewer.
> > > >
> > > > [1] Zarlenga, Mateo Espinosa et al. “Concept Embedding Models.” ArXiv abs/2209.09056 (2022): n. pag.

---

### Official Review · Reviewer_LQqh · 2024-10-31

**Soundness:** 2
**Presentation:** 1
**Contribution:** 2
**Rating:** 6
**Confidence:** 4

**Summary:**

Over the past few years, there has been a lot of interest in designing models that are interpretable by construction. Concept Bottleneck Models are one such approach; they first predict an intermediate set of human-specified concepts and then use the concepts to predict the target. These intermediate concepts allow humans to understand the dynamics of model predictions through interventions. One key drawback of such an approach is that they assume each concept independently influences the predictions, while in practice, there is always a complex relationship between the concepts and how they influence the target variable. The paper addresses the above issue by introducing the Causal Graph Concept Graph Model, which is transparent by design and captures the complex dynamics of a model's decision processes.

**Strengths:**

A serious shortcoming of CBMs is that they assume that each human-specified concept independently influences the model's predictions. The authors address this issue by proposing a Causal Graph Concept model that captures the intricacies of the dependencies between the concepts & the target variable and among them.

As the proposed model is causally transparent by design, it allows a human to understand the inner workings of the model by enabling them to ask questions, such as:
- What is the relationship between feature x & label y?
- What happens to the prediction if I set feature x to k?
- What if feature x was k' instead of k?
The proposed model's performance matches or is within the ballpark of the opaque state-of-the-art models on the chosen set of benchmarks.

Theoretically, it also allows humans to intervene and correct erroneous intermediate reasoning steps.

**Weaknesses:**

- The paper isn't written well, and some sections are hard to follow.
- One of the key weaknesses lies in the experiments.
     - Why only choose simpler benchmarks, like CelebA & CIFAR10, and benchmarks with semantically richer tasks like AwA2, CUB, etc.?
     - Understanding what the authors mean by joint training is a bit hard.
          - Are the concept encoders also trained along with the process of learning CGMs?
          - What formulation of CBMs do the authors build on? (independent bottleneck, joint bottleneck, or sequential bottleneck?)
          - What is the architecture of the bottleneck layer? Does it influence the quality of CGMs learnt?
     - One contribution is that CGMs allow human-in-the-loop corrections of erroneous intermediate reasoning steps. I don't see any experiments to support this. Did you evaluate whether the learnt causal graphs are sound and valuable? Did you do a user study to assess their utility to users?
     - Do you have any intuition about why there is a drop in Figures 4 and 5 for a certain number of intervened concepts?
     - Column 4 is Table 3 is CIFAR10?
     - Line 345 says CIFAR10 is an animal classification dataset. Did you use the entire dataset or a subset for evaluation?
- Another issue with CBMs is that they assume that human-specified concepts are readily available, which rarely happens in practice. Much work has been done in the literature to relax this assumption. I wonder why the authors build on the vanilla version of CBMs.

**Questions:**

Refer to Weaknesses

---

> ### Author Response · Authors · 2024-11-22
> **Response to Reviewer LQqh**
>
> ### Some sections are hard to follow
> Thank you so much for pointing this out. We want this paper to be as clear as possible so this feedback is helpful. With this in mind, could you please point us to specific sentences or sections so that we can improve them?
>
> ---
>
> ### Simple benchmarks
> **CelebA is a standard benchmark in CBM papers (comparable to CUB), and we used CIFAR10 for a challenging real-world condition where we assumed that concept labels were missing in the dataset.** Besides, our contributions extend beyond empirical evaluations by providing a theoretically motivated novel framework and a new way of thinking about causal opacity within the popular context of concept-based XAI.
>
> ---
>
> ### Joint training
> **Joint training is a method for training Concept Bottleneck Models**. To clarify this point, we added the following in App. G.2: “In our experiments, all baselines are trained using a joint training technique commonly used in CBMs [1]. In joint training, the model is trained end-to-end with the task predictor directly taking the concept encoder’s outputs as input, enabling simultaneous optimization of both concept and task predictions.”
>
> ---
>
> ### What is the architecture of the bottleneck layer? Does it influence the quality of CGMs learnt?
> **The details of the architecture of the bottleneck layer are described App D**. To better explain this in the main text, we moved the key component of the bottleneck layer from App. D after Sec 3.3. To support this choice we conducted an ablation study replacing high-dimensional embeddings with scalar values and showed how this affected performance and interpretability (see shared answer “Endogenous variable representation: scalar values vs. high-dimensional embeddings”).
>
> ---
>
> ### Missing experiments on human-in-the-loop corrections of the intermediate reasoning steps
> **The results of these experiments are discussed in the second and fourth paragraph of section 4.1 and shown in Fig 4, 5, 12, 13, and in Table 3.**
>
> ---
>
> ### Check whether the learnt causal graphs are sound and valuable
> **The learnt causal graphs enable the model to match the performance of black box models (valuable) and pick up known biases in the data (sound)**. The generated causal graphs represent the actual reasoning path of the model, which is typically unknown when considering a black box model. If a user values knowing the reasoning path of the model, then these graphs could be useful. For instance, we show how the generated graphs enable users to de-bias the model by blocking all paths between two variables (results reported in Table 3). For example, if a user does not want “gender” to be a cause of “attractiveness” in CelebA, the user can apply do-interventions that remove all edges connecting “gender” to “attractiveness”. This way, if we change the prediction for “gender” (formerly a cause), this can no longer have any effect on “attractiveness”. However, we  clarify that we did not run a user study as that went beyond the scope of this current work and our contributions.
>
> ---
>
> ### Do you have any intuition about why there is a drop in Figures 4 and 5 for a certain number of intervened concepts?
>  **A drop in performance between intervention $i$ and intervention $i+1$ indicates that the label intervened at step $i$ was already well-predicted.** Figures 4 and 5 show the delta in performance before and after interventions on **all non-intervened** labels/concepts. Consequently, removing this well-predicted concept from the computation had a negative impact, outweighing the benefits of the intervention. If a label is correctly predicted, excluding it from the metric calculation will reduce the delta score.
>
> ---
>
> ### Line 345 says CIFAR10 is an animal classification dataset. Did you use the entire dataset or a subset for evaluation?
> **Enitre dataset**. Thanks for pointing out our error in describing the data! We will replace it with “image classification dataset”
>
> ---
>
> [1] Koh, Pang Wei et al. “Concept Bottleneck Models.” ArXiv abs/2007.04612 (2020): n. pag.

---

> > ### Comment · Reviewer_LQqh · 2024-11-23
> >
> > Dear Authors,
> >
> > I appreciate the time and effort invested in providing detailed clarifications that address my and other reviewers' concerns (I will update my scores). I found section 3 to be challenging to follow (a suggestion would be, in addition to the theory, to add some intuition regarding its implications).
> >
> > I agree that the paper also makes theoretical contributions, but incorporating datasets like CUB and AwA2 would strengthen it further.

---

> > > ### Author Response · Authors · 2024-12-04
> > >
> > > We thank the reviewer for pointing this out.
> > >
> > > To address this, we will include a detailed example in the camera-ready version to illustrate all the steps in Section 3. Specifically, we will build on the example introduced in the introduction (heart attack ← smoking ← lung cancer) to provide a more intuitive understanding of the theory and its implications. We hope this addition will make each step clearer.

---

### Official Review · Reviewer_hQg6 · 2024-11-04

**Soundness:** 3
**Presentation:** 3
**Contribution:** 3
**Rating:** 8
**Confidence:** 2

**Summary:**

The authors propose a new class of interpretable models whose decision-making process is causally transparent by design. The method represents relations between high-level concepts, associated with each data point, as a DAG allowing for interventional and counterfactual analysis of predictions and human improvements during the inference process.

The authors demonstrate that their approach matches the generalization performance of SoTA models. Additional experimental results showcase how explicit causal representation can be efficiently used to increase interpretability and improve performance with human interventions.

**Strengths:**

1. The paper is cleanly written.
2. The approach is well-placed in the literature.
3. The experimental section is extensive.
4. The method description is detailed and well-structured.

**Weaknesses:**

1. One bit of experimental evaluation is unclear to me (see Questions)

**Questions:**

1. I do not understand the experiment described in the last paragraph of the experimental section (lines 463-476).  Could the authors please provide an example? How are variables i and j selected for each dataset? Are the results in Table 3. averaged over all possible pairs? How do the authors decide which variables to intervene on (i.e. what structure is used as Ground Truth)? Why is 0 a desirable outcome?  - the authors write that the aim of this procedure is debiasing, to my understanding 0 RCCE means no change.
2. Table 3 has two columns titled “CELEBA”

---

> ### Author Response · Authors · 2024-11-22
> **Response to Reviewer hQg6**
>
> ### "I do not understand the experiment described in the last paragraph of the experimental section (lines 463-476). Could the authors please provide an example? How are variables $i$ and $j$ selected for each dataset? Are the results in Table 3. averaged over all possible pairs? How do the authors decide which variables to intervene on (i.e. what structure is used as Ground Truth)?"
> **The last experiment aims to demonstrate that our model can be effectively debiased through do-interventions.** The idea behind this procedure is described in the paragraph “Model verification and blocking”. However, to give more details about the choice of the variables, we added the following description in App. H.7:
>
> 1. Selection of Variables $i$ and $j$:
> We randomly select a pair of concepts, $i$ (parent) and $j$ (child), connected in the graph discovered by the CD algorithm or CausalCGM.
>
> 2. Do-Intervention:
> A do-intervention is applied to the child node ($j$) to cut off its dependence on the parent node ($i$). In our model, this intervention ensures that all subsequent nodes in the graph become independent of the parent node, as information flow is blocked by design.
>
> 3. Measuring the Impact:
> After the intervention, we modify the value of the parent node ($i$) and measure its impact on the final label (leaf node) with $i$ as an ancestor. If no effect is observed, it indicates successful debiasing, as expected by our model’s design. Cutting the edge breaks the information flow through the graph—from root to leaf—preventing the parent node from influencing subsequent nodes.
>
> 4. Comparison with Baselines:
> Baselines have no graph constraints, therefore the do-intervention modifies the value of a concept but does not alter the decision-making process. As a result, the parent node’s influence cannot be entirely removed. This contrasts our model’s ability to block the information flow fully.
>
> ---
>
> ### Why is 0 a desirable outcome? (Table 3)
> Consider a graph $A \rightarrow B \rightarrow C$. For this graph, the variable $A$ is a cause of the variable $C$. As a result, we expect the average causal effect of $A$ on $C$ to be higher than $0$. Our experiment aims to show that intervening on $B$ makes $A$ and $C$ causally independent (cf. with L323-331). To show this, we apply a do-intervention in $B$, removing the edge $A \rightarrow B$ and generating two disconnected graphs: $A$ and $B \rightarrow C$. Now, we expect that the residual average causal effect of $A$ on $C$ is $0$ because there is no longer a connection between $A$ and $C$. As a result, $0$ is the optimal desirable outcome. To improve clarity, we added this example in App. H.7.

---

> > ### Comment · Reviewer_hQg6 · 2024-11-29
> >
> > I appreciate the effort the authors have put into preparing the rebuttal. Thank you for addressing my questions.

---

### Official Review · Reviewer_eiBf · 2024-11-04

**Soundness:** 1
**Presentation:** 2
**Contribution:** 2
**Rating:** 6
**Confidence:** 3

**Summary:**

In their work, the authors construct causal concept bottleneck models (CCBMs), which are interpretable models that aim to reveal the underlying decision process of deep learning models. The models are built with the assumptions that all relevant human-aligned concepts are known a priori and that these concepts are direct causes of the class prediction. The authors propose measure the predictive power of concepts towards each other other to figure out variable parents. The authors leverage the assumption, that distribution predictions between features predicted by noise, and features predicted from the prior noise-predicted values are invariant. The recovered structure is then used to extract logic rules and trace-back decision.

The approach is evaluated on smaller image datasets, including CelebA and CIFAR 10, showing performance gains over other non-causal methods.


[Disclaimer: This is an emergency review]

**Strengths:**

The authors extend prior approaches by dropping the restriction of causally independent concepts and trying to approximate the underlying true graphical causal relation. This is an important extension, as most real-world settings, feature complex interactions between concepts and allows to inspect the consequences of interventions on the modeled process.

By observing predictive power among variables, the authors are able to (partially) recover the individual parents of the variables. If trained correctly, CCGM are therefore deep models, which are interpretable by design and possibly allow for human-in-the-loop adjustments.

The presented results improve over prior CBM and CEM, while lacking only slightly behind 'causally opaque' black-box models. The authors show good predictive performance under interventions, and are able to extract some underlying logic rules from the learned models.

**Weaknesses:**

The weaknesses concern extend to which the presented method 'is actually causal'. While existing works in the field of causality are mainly concerned with providing identifiability results, the presented method mainly leverages predictive power to identify causal relations between variables, which might not coincide with learning the correct causal structure and lead the model to learn spurious associations.

1) Measuring the predictive performance of features towards each other does guarantee to recovery the underlying causal structure. Generally, additional assumptions such as interventions or sufficient variability need to be assumed. In that regard, theorem 3.2 does not state any of these assumptions and is provided without proof.

2) In that regard the paper lacks a comparison and detailed discussion to existing methods in the field of causal representation learning [1], such as differences to other existing deep causal approaches ([2-4]), and -with regard to the prior point- guarantees on the identifiability of (latent) concepts [5-7].
3) Variables $v'_i$ are predicted just from their exogenous noise variables $u_i$. Assuming that, variables are indeed embedded into a causal structure, and assuming that at least for some $v_i$ the parent set is non-empty, it can not be that $p(v'_i\mid pa_{V'}(v_i),u_i) = p(v_i \mid u_i)$.
4) The experimental section compares to only non-causal methods (CEM and CBM) on their predictive performance (accuracy). Given the above discussion, it is unclear whether the models have actually learned the correspond process. No analysis or discussion towards identifiability and possible avoidance of confounding is presented. As a result a rather strong drop-off in performance between CCGM+CD (recovering the graph from ground-truth labels) and CCGM is observed. In particular, when the causal graph is recovered from data, the presented causal CGM+CD approach might by very similar to Neural Causal Models (NCM) [4].



**Minor**

* Table 1 is hard to read. I suggest breaking "semantic transparency" and "causal transparency" into two lines to make the texts bigger.


[1] Schölkopf, Bernhard, et al. "Toward causal representation learning." *Proceedings of the IEEE* 109.5 (2021): 612-634.
[2] Kladny, Klaus-Rudolf, et al. "Deep backtracking counterfactuals for causally compliant explanations." *arXiv preprint arXiv:2310.07665* (2023).
[3] Lippe, Phillip, et al. "Biscuit: Causal representation learning from binary interactions." *Uncertainty in Artificial Intelligence*. PMLR, 2023.
[4] Xia, Kevin, Yushu Pan, and Elias Bareinboim. "Neural causal models for counterfactual identification and estimation." *arXiv preprint arXiv:2210.00035* (2022).
[5] Brehmer, Johann, et al. "Weakly supervised causal representation learning." *Advances in Neural Information Processing Systems* 35 (2022): 38319-38331.
[6] Wendong, Liang, et al. "Causal component analysis." *Advances in Neural Information Processing Systems* 36 (2024).
[7] von Kügelgen, Julius, et al. "Nonparametric identifiability of causal representations from unknown interventions." *Advances in Neural Information Processing Systems* 36 (2024).

**Questions:**

My questions mainly concern the weaknesses mentioned above. I would kindly like to ask the authors the following questions:

1) How does method guarantee identifiability and non-confoundedness of concept predictions. Could the authors elaborate on the implications of theorem 3.2 towards this question and/or highlight possible limitations that might arise?
2) Regarding point 3: Could the further explain how distributions of Theorem 3.2 can be equal, given, that $v'_j$ are inferred from exogenous variables only?
3) How, does the presented approach compare to other causal deep-learning methods? Could the authors present further insights or explain, why other methods might not be comparable?

---

> ### Author Response · Authors · 2024-11-22
> **Response to Reviewer eiBf (1/2)**
>
> We greatly appreciate the reviewer’s thoughtful comments and insights. We would like to kindly point out that several of the concerns raised appear to stem from evaluating our approach as a causal discovery/identification/estimation method (all the referenced papers are on this research area). We respectfully invite the reviewer to revisit these points with the clarification that our model is explicitly *not* intended as a causal discovery/identification/estimation method but as a solution to the causal opacity issue, as we emphasized extensively in the introductory section of the manuscript (L38), in the main body (Remark 3.3), and in the appendix (App. J). The differing nature of the target problem (causal opacity vs. causal discovery/identification/estimation) has a significant impact on objectives, constraints, optimization processes, and scalability, as discussed in the following answers.
>
> ---
>
> ### Theorem 3.2 is provided without proof
> **We include the proof of theorem 3.2  in App. C whilst discussing the main intuition in the main body of the text (as stated in L198).**
>
> ---
>
> ### Additional assumptions, such as interventions or sufficient variability, need to be assumed to guarantee the recovery of the underlying causal structure. In that regard, theorem 3.2 does not state any of these assumptions
> **Theorem 3.2 does not require any of these assumptions as the problem of causal opacity is not to recover the causal mechanisms in the real world (cf. to shared answer on identifiability).** In particular, addressing causal opacity does not pose any constraints in terms of interventions or sufficient variability in observations.
>
> ---
>
> ### Guarantee non-confoundness
> **1. Causal CGM does not guarantee non-confoundness just as it does not guarantee identifiability. Please see our shared answer on identifiability for our argument for why this is a feature and not a bug.**
>
> **2. Our results (Fig 4, 5, 12, 13) show that interventions are still more effective w.r.t. existing CBMs despite the presence of potential confounders.**
>
> ---
>
> ### Variables are predicted just from their exogenous noise variables
> **1. We would like to point out that variables are not just predicted from their exogenous variables alone. As we show in Def. 3.1 and Theorem 3.2 (and more in detail in App. D), endogenous variables are predicted from both their endogenous copies and their exogenous variables.** In those statements, $\mathcal{V}$ and$ \mathcal{V}^\prime$ are endogenous variables satisfying $p(v_i \mid pa_{\mathcal{V}'}(v_i), u_i; \theta_f) = p(v_i \mid pa_{\mathcal{V}}(v_i), u_i; \theta_f)$
>
> **2. As evidence that indeed variables are not just predicted from their exogenous noise variables, we would like to point out that our intervention experiments show that endogenous ancestors have a causal effect on endogenous descendants (cf. with Fig 4, 5, 12, 13)**. This would not be possible unless there is a dependence on endogenous variables too.
>
> ---
>
> ### Performance gap between CCGM+CD and CCGM
> **1. We did not observe any significant gap in task accuracy (Tab 1).**  So, we assume the reviewer refers to the gap observed in Fig 4 and 5 representing intervention effectiveness.
>
> **2. CCGM+CD shows that combining causal representation learning with our method can improve intervention effectiveness.** This also highlights why our method is neither a causal representation learning technique nor a causal discovery technique. Rather, the proposed approach is complementary and can be combined with such techniques.
>
> **3. The gap in intervention effectiveness is a form of explanation.** This gap is the result of the intended use of our model. It shows that the causal structure used by the model for inference might differ from a human’s causal structure and the data-generating process’ causal structure. As a result, interventions may not end up having the intended effect.
>
> **4. When CCGM+CD does not scale (Tab 1) because of the CD component (see CIFAR10), CCGM becomes the preferred solution for interventions (more effective than existing CBMs).**

---

> > ### Author Response · Authors · 2024-11-22
> > **Response to Reviewer eiBf (2/2)**
> >
> > ### When the causal graph is recovered from data, the presented causal CGM+CD approach might be very similar to Neural Causal Models (NCM)
> > **1. Causal CGMs and NCMs address distinct problems: causal opacity (Causal CGMs) versus causal identification and estimation (NCMs).** The differing objectives of these methodologies have a significant impact on their constraints, optimization processes, and scalability.
> >
> > **2. Causal identification constraints limit the scalability of NCMs to simple, synthetic experimental settings.** The NCM architecture and training algorithm are heavily constrained (see, for example, Appendix C.2 in the original NCM paper [1]). As a result, NCMs are typically evaluated on small synthetic datasets with very few variables (up to 4). This limitation is evident in the experimental results reported in the original NCM paper [1], Figure 4, and in the more recent paper referenced in the review [2], Figure 4.
> >
> > **3. In their original formulation, NCMs are restricted to structured data.** In contrast, Causal CGMs, similar to Concept Bottleneck Models, are designed to maintain causal transparency even when applied to unstructured data (e.g., images).
> >
> > **4. The architectures of NCMs and Causal CGMs differ fundamentally in the endogenous variable representation.**  NCMs represent endogenous variables as scalars, whereas Causal CGMs use high-dimensional embeddings (as detailed in Appendix D). These embeddings enable Causal CGMs to match classification performance of black-boxes in more realistic settings (as shown in Table 1). To better highlight this architectural property, we have moved Equation 13 to follow Section 3.3 and provided a more detailed explanation of this aspect (refer to the shared response ”Endogenous variable representation: scalar values vs. high-dimensional embeddings”).
> >
> > ----
> >
> > [1] Xia, Kevin, et al. "The causal-neural connection: Expressiveness, learnability, and inference." Advances in Neural Information Processing Systems 34 (2021): 10823-10836.
> >
> > [2] Xia, Kevin, Yushu Pan, and Elias Bareinboim. "Neural causal models for counterfactual identification and estimation." arXiv preprint arXiv:2210.00035 (2022).

---

> > > ### Comment · Reviewer_eiBf · 2024-11-24
> > >
> > > Dear Authors,
> > > thank you for answering my questions comprehensively. Given the rather short time frame of the emergency review, I seem to have missed some of the already discussed relations to CRL and causal discovery in the Appendix. I appreciate the added clarifications to the introduction and in the appendix that better distinguish differences to CRL and CD and highlight the relation to model identifiability in this paper.
> > >
> > > I generally agree with the more relaxed conditions when considering causal opacity as it concerns the model's inherent decision making process. However, given that the role of causal opacity is to introspect and explain how particular decisions of a model come to be, I would like to recommend the authors to at least discuss desirable properties of learned models. In particular, I find the following statement "As a result, there is no single 'ground truth' causal structure to identify" in Appendix L to be rather uninformative, and having "models can learn different representations to predict the same outputs" (Appendix L), might in particular not hold for interventional and counterfactual queries (even w.r.t. different models trained on the same data). While the ability of correctly identifying the 'correct' causal graph might only be possible under certain conditions, up to some particular equivalence class (e.g. Markov Equivalence) and is the general concern of CRL and CD, I would like to recommend to the authors to still include the fact that the desired property of a faithful model in terms of predictive power is likely to coincide with it resembling the 'true' underlying graph to guarantee the correct estimation of interventional (and CF) queries.
> > >
> > > Nonetheless, –and considering answers to the other reviewers– I appreciate the made adjustments to the paper and have increased my score accordingly.
> > >
> > > Best Regards,

---

> > > > ### Author Response · Authors · 2024-12-03
> > > >
> > > > We would like to thank the reviewer again for the comment.
> > > >
> > > > However, we would like to further clarify these last point.
> > > >
> > > > > I would like to recommend to the authors to still include the fact that the desired property of a faithful model in terms of predictive power is likely to coincide with it resembling the 'true' underlying graph to guarantee the correct estimation of interventional (and CF) queries.
> > > >
> > > > We strongly agree with the reviewer on this point. The experiment we defined using Causal CGM + CD was specifically designed with this intent in mind.
> > > >
> > > > To address this valuable observation, we will add the following sentence at line L370 in the camera-ready version:
> > > > "This approach highlights that the desired property of a faithful model in terms of predictive power is likely to coincide with its resemblance to the ‘true’ underlying graph, as this alignment ensures accurate estimation of interventional and counterfactual queries about the data generating process."

---

### Official Review · Reviewer_ytSc · 2024-11-09

**Soundness:** 3
**Presentation:** 3
**Contribution:** 3
**Rating:** 8
**Confidence:** 2

**Summary:**

This paper presents Causal Concept Graph Models (Causal CGMs), a concept-based architecture whose decision-making process is causally transparent by design. It avoids the unrealistic assumption that concepts must be causally independent and direct causes of the class prediction. The paper aims to design Deep Learning  models where each prediction can be traced back to a chain of
semantically meaningful causes.

**Strengths:**

-  The paper is well written.
-  The idea of introducing causal transparency in concept-based models is interesting.
-  The experiments are sound.

**Weaknesses:**

-  > *Causal opacity refers to the difficulty in understanding the mechanisms behind a model’s decision-making process.*

  The paper frequently references the concept of causal opacity, but it is too vaguely defined to be fully understood.

- How the model performs abduction to answer counterfactual queries is unclear.

- Minor: The description of Figure 3 is inconsistent with the picture.

- It seems the proposed model sacrifices accuracy to improve interpretability.

**Questions:**

- > Concept Graph Models duplicate the layer of interpretable variables $V$ by introducing an additional layer of identical copies $V^{′}$ . Using this additional layer, Causal CGMs make predictions of each $v_{i} \in V$ using as possible inputs all $v_{j}' \in V^{′}$ for all $j \neq i$

How do you guarantee the causal graph will be acyclic?

---

> ### Author Response · Authors · 2024-11-22
> **Response to Reviewer ytSc**
>
> ### The paper frequently references the concept of causal opacity, but it is too vaguely defined to be fully understood.
> **Causal opacity can be understood using Pearl's framework of causality (cf with [1], L29-43 and App. A).** One way to define causal opacity is to consider a similar problem: causal discovery. While causal discovery examines how variables are causally related in the real world, causal opacity focuses on understanding how variables are causally related in a model's decision-making process. The object under study (the “world” in causal discovery and the “model” in causal opacity) is different, but the ladder of causality is the same. To further clarify this, we modified two sentences in the introduction: (L31) “... Similarly to causal discovery, causal opacity can be assessed …”; (L36): “... instead of k?”). However, as opposed to causal discovery, the target of causal opacity is understanding …”.
>
> ----
>
> ### How the model performs abduction to answer counterfactual queries is unclear
> **CCGMs follow [2] performing abduction by inferring the value of exogenous variables based on the available evidence (L308-10).** To further clarify this, we will add the following in App. D where we discuss the details of the architecture: “In causal CGMs we use the exogenous encoder to predict the value of exogenous variables as we consider that the observed input (e.g., an image’s pixels) holds most of the contextual variability required to infer exogenous variables. For instance, an image provides information about background, lighting conditions, and object locations. All this information is used to anchor the endogenous predictor to a specific context and to correctly infer the values of the exogenous variables and the endogenous roots of the causal graph.”
>
> ---
>
> ### It seems the proposed model sacrifices accuracy to improve interpretability
> **1. Our primary goal was to enhance interpretability with minimal accuracy loss, not to surpass the accuracy of black-box models (L358-62).** Causal CGM’s enhanced interpretability enables users to intervene and debias the model at inference time, as opposed to black boxes and more efficiently than existing CBMs. We point out that in several critical instances where verification is necessary (e.g., hiring, law, loan management), this makes our models plausible options for these tasks as they still have an accuracy close to that of black-box whilst being fully interpretable. In contrast, black-box models would not satisfy the verifiability requirements.
>
> **2. In our experiments, the accuracy drop is at most 1.5 p.p. w.r.t. a black box model (Tab 1)**. While the trade-off between accuracy and interpretability is well-known, our model achieves comparable performance to black-box models while providing an interpretable decision-making process.
>
> ----
>
> ### How do you guarantee the causal graph will be acyclic?
> **1. Eq 4 is used in the loss function to make the graph acyclic.**
>
> **2. After training, we guarantee acyclicity by removing the least important edges that create cycles if any**. Thanks for pointing this out. To clarify our statement,  We will include the following sentence in Appendix E: “During training, we optimize a loss that encourages the model to construct a DAG. However, in rare cases, some edges with very small weights may still make the adjacency matrix $A$ cyclic. At inference time, we address this by iteratively removing the edges with the smallest weights from the cycles until the graph becomes a DAG.”
>
> ----
>
> [1] Facchini, Alessandro, and Alberto Termine. "Towards a Taxonomy for the Opacity of AI Systems." Conference on philosophy and theory of artificial intelligence. Cham: Springer International Publishing, 2021.
>
> [2] Pearl, Judea. Causal inference in statistics: a primer. John Wiley & Sons, 2016.

---

> > ### Comment · Reviewer_ytSc · 2024-11-27
> >
> > Thank you very much for your clarifications.

---

### Author Response · Authors · 2024-11-22
**Answer to all reviewers and ACs (1/2)**

We thank the reviewers for their insightful feedback. We are glad to read that reviewers found our work to be “important” (eiBf), “well written” (ytSc, hQg6, JdcH), “well-placed in the literature” (hQg6), with “sound” (ytSc) and “extensive” (hQg6) experiments that “(somewhat surprisingly) show that causal CGMs perform similarly to black box models that do not enforce a causal bottleneck” (JdcH).

We are thankful for all the suggestions the reviewers made. These have certainly improved the quality of our manuscript and we hope we can address your concerns here.

Below, we reply to questions shared by two or more reviewers.  We reply to specific questions reviewers had in comments under their respective reviews.

## Summary of Changes
In our revised manuscript, we have included the following changes (in purple) to take into account the reviewers’ feedback:
- Introduction: clarified the role of causal opacity and the relation with causal discovery
- New sec 3.4: moved part of App. D in the main text to highlight a key difference wrt existing methods which is represented by the use of high-dimensional representations for endogenous variables
- App. D: clarified the role of the exogenous encoder
- App. E: clarified how we make the graph acyclic in corner cases
- App G.2: added explanation for joint training procedure used in the CBM literature
- App H.7: added appendix H.7 to provide more details on the debiasing experiment
- App L: added an example showing why identifiability is not relevant in the context of causal opacity

## Shared Answers

### Assumption that human-specified concepts are readily available
**Causal CGMs do not make this assumption, as shown in the CIFAR10 experiment where concepts are obtained using a label-free technique (L345-6)**. We approached this following the label-free method in [1] and discussed it in Section 4.

---

### The discovered causal structure may not match the “true” causal structure of the data generating process
**Causal opacity (and the related form of causal understanding) neither implies nor requires discovering the “true” causal structure of the data-generating process**. We argue that this is desirable as it enables us to capture the model’s causal reasoning mechanisms when they may differ from the ones involved in the data-generating process. We explain this key point in the introduction (L35-43), when introducing the problem addressed by our method (L138-140), in remark 3.3 (L228-234), and in App. J. We added the following sentence in the introduction of the updated manuscript: “Causal opacity (and the related form of causal understanding) neither implies nor requires discovering the “true” causal structure of the data-generating process”.

---

### Guarantee identifiability
**CGMs causal structure is not identifiable: this is a feature and not a bug, as discussed in Remark 3.3 and App. L.** To explain this, in our paper we use an analogy between causal discovery and causal opacity. While causal discovery examines how variables are causally related in the real world, causal opacity focuses on understanding how variables are causally related in a model's decision-making process. Although the problems are similar, the subjects differ—causal discovery targets the world's causal structure, whereas causal opacity studies the causal structure of a decision-making process. This distinction impacts key properties, including identifiability. To further clarify this point we will add the following example in App  L: *“In the world, there is a single ground truth causal structure (e.g., “smoking causes lung cancer”). Guaranteeing identifiability is crucial here because it ensures that we can reliably recover the ground truth causal graph. In contrast, causal opacity deals with DL models, which are systems that can implement a variety of decision-making mechanisms (e.g., a valid decision mechanism for a DL model is: “the model predicts that an individual has lung cancer and this causes the model to predict that the individual has a smoking habit”). These mechanisms may differ even when trained on the same data, as DL models can learn different representations to predict the same outputs. As a result, there is no single "ground truth" causal structure to identify—multiple valid causal graphs exist, each reflecting a different way the model processes information. This inherent variability makes identifiability irrelevant in the context of causal opacity.”* Instead, the focus shifts to evaluating and comparing the causal structures across different DL models to verify that they align with desirable properties, such as fairness.

---

[1] Oikarinen, Tuomas P. et al. “Label-Free Concept Bottleneck Models.” ICLR (2023).

---

> ### Author Response · Authors · 2024-11-22
> **Answer to all reviewers and ACs (2/2)**
>
> ### Relations and comparison with causal representation learning
> **Causal CGMs are not an example of CRL, as discussed in Remark 3.3 and App. K.** We agree with this statement. In fact, we remark this in our experiments by showing how CCGM and causal representation learning are complementary; they are not competitors. CCGMs can be combined with any existing causal representation learning method. In the experiments we call this combination “CCGM+CD”, representing the combination of CCGM with causal discovery methods to improve the effectiveness of interventions.
>
> ----
>
> ### Endogenous variable representation: scalar values vs. high-dimensional embeddings
> **Unlike existing methods (e.g., Neural Causal Models and Causal VAEs), the Causal CGMs architecture is built using expressive high-dimensional representations for endogenous variables.** Details of this architectural aspect are provided in Appendix D, and its impact on the predictive performance of Causal CGMs is discussed in Section 4.1 (Line 371). However, as this is a significant difference w.r.t. existing methods, such as Neural Causal Models and Causal VAEs, we emphasized this distinction even further in Section 3 by expanding the description provided in Appendix D as follows:
> “As observed by [2], using scalar representations for concepts (corresponding to endogenous copies in our context) can significantly degrade predictive performance in realistic settings. To address this issue, and inspired by [3], Causal CGMs employ high-dimensional representations for endogenous copies $\mathbf{v}_j'$. For each endogenous copy, Causal CGMs learn a mixture of two embeddings with explicit semantics, representing the variable's state. This design enables the model to construct evidence both in favor of and against a variable's state and supports simple concept interventions, as it allows switching between the two embedding states during interventions. Specifically, we represent the context of each variable with two embeddings: $[\mathbf{c}_j^+, \mathbf{c}_j^-] = \mathbf{u}_j = \zeta(\mathbf{x}), \quad \mathbf{c}_j^+, \mathbf{c}_j^- \in \mathbb{R}^m$. Each embedding carries specific semantics: $\mathbf{c}_j^+$ represents the active state of the variable (\texttt{true}), while $\mathbf{c}_j^-$ represents the inactive state (\texttt{false}). Once these semantic embeddings are computed, the final endogenous embedding $\mathbf{v}_j'$ for $v_j$ is constructed as a mixture of $\mathbf{c}_j^+$ and $\mathbf{c}_j^-$, weighted by the endogenous state:
> \begin{equation}
> 	\mathbf{v}_j' = v_j' \mathbf{c}_j^+ + (1 - v_j') \mathbf{c}_j^-
> \end{equation}
> This formulation serves two primary purposes: (i) it forces the model to rely exclusively on $\mathbf{c}_j^+$ when the $j$-th endogenous variable is active ($v_j’ = 1$) and on $\mathbf{c}_j^-$ when inactive, thereby creating two distinct and semantically meaningful latent spaces; (ii) It enables a straightforward intervention strategy, where one can switch between embedding states to correct a mispredicted endogenous variable.”
>
> Additionally, we conducted an ablation study by replacing embeddings with scalar values in Causal CGM following a CBM-like approach on CelebA. The results demonstrate that when following a causal graph, this approach struggles to achieve the same performance as other models due to its limited expressiveness ($72.72 \pm 3.70$ compared to $78.42 \pm 0.42$ achieved with embeddings).
> However, when the model is allowed to learn the graph, this approach achieves performance comparable to other Causal CGM or CBM models ($78.64 \pm 0.30$). Notably, the learned graph in this case only includes edges from concepts to labels, effectively reducing it to a CBM structure. As shown in other experiments, this structure is suboptimal due to its reduced impact in interventions and the inability to effectively debias the model.
>
> ----
>
> ### Column 4 is Table 3 is CIFAR10?
> **Yes**. We thank the reviewers for noticing this. We have fixed this typo in our updated manuscript.
>
> ----
>
> [2] Mahinpei, Anita, et al. "Promises and pitfalls of black-box concept learning models." arXiv preprint arXiv:2106.13314 (2021).
>
> [3] Espinosa Zarlenga, Mateo, et al. "Concept embedding models: Beyond the accuracy-explainability trade-off." Advances in Neural Information Processing Systems 35 (2022): 21400-21413.

---

### Author Response · Authors · 2024-12-03
**Thanks**

We would like to thank all the reviewers for the comments and for helping us improve the paper.

---

### Meta-Review · Area_Chair_dsUh · 2024-12-20

**Metareview:**

The paper shows how to construct causal concept bottleneck models (CCBMs), i.e., interpretable models that aim to reveal the underlying decision process of deep learning models.  Overall, the reviewers lean towards acceptance, and I agree. While the datasets considered are rather small, the results are interesting as they pave the way to deep models where each prediction can be traced back to a chain of semantically meaningful causes. The authors, however, should spend more time on explaining key terms of the approach such as causal opacity,

**Additional Comments On Reviewer Discussion:**

The discussion arose from issues raised in the reviews and led to increased scores. One emergency reviewer acknowledged that the review missed some information provided in the appendix. The discussion also touched upon missing datasets. Overall, it helped to arrive at the overall decision.

---

### Decision · Program_Chairs · 2025-01-22

Accept (Poster)